# GEM: A Dynamic Tracking Model for Mesoscale Eddies in the Ocean

Qiu-Yang Li[1], Liang Sun[1, 2], Sheng-Fu Lin[1]

[1]School of Earth and Space Sciences, University of Science and Technology of China, 230026, Hefei, China.

[2]State Key Laboratory of Satellite Ocean Environment Dynamics, Second Institute of Oceanography, State Oceanic Administration, Hangzhou, 310012, PR China.

*Correspondence to*: L. Sun (sunl@ustc.edu.cn)

**Abstract**

Genealogical Evolution Model (GEM) is an efficient logical model used to track dynamic evolution of mesoscale eddies in the ocean. It can distinguish different dynamic processes (e.g., merging and splitting) within a dynamic evolution pattern, which is difficult to accomplish using other tracking methods. To this end, GEM first uses a two-dimensional (2-D) similarity vector (i.e. a pair of ratios of overlap area between two eddies to the area of each eddy) rather than a scalar to measure the similarity between eddies, which effectively solves the ''missing eddy" problem (temporarily lost eddy in tracking). Second, GEM uses both parents (a new eddy) and children (e.g., splitting eddies from parent eddy) in tracking, and the dynamic processes are described as birth and death of different generations. Additionally, a look-ahead approach with selection rules effectively simplifies computation and recording. All of the computational steps are linear and do not include iteration. Given the pixel number of the target region $L$, the maximum number of eddies $M$, the number $N$ of look-ahead time steps, and the total number of time steps $T$, the total time complexity is $O(LM(N+1)T)$. The tracking of each eddy is very smooth because we require that the snapshots of each eddy on adjacent days overlap one another.

Although eddy splitting or merging is ubiquitous in the ocean, they have different geographic distribution in the Northern Pacific Ocean. Both the merging and splitting rates of the eddies are high, especially at the western boundary, in currents and in "eddy deserts". GEM is useful not only for satellite-based observational data but also for numerical simulation outputs. It is potentially useful for studying dynamic processes in other related fields, e.g., the dynamics of cyclones in meteorology.

## 1  Introduction

Eddies are ubiquitous in the ocean, and they move from one place to another [Chelton and Schlax, 1996; Chelton et al., 2007]. Eddies in the ocean can cause large-scale transports of heat, salt and other tracers [Bennett and White, 1986; Chelton et al., 2011a; Dong et al., 2014; McGillicuddy et al., 2011] by trapping these passive tracers inside the eddies. Such transports may have important impacts on the environment and climate of the ocean [Dong et al., 2014]. To address various applications in the studies that use satellite products of sea level anomaly (SLA) data [e.g., Chelton et al., 2011b] and numerical simulation outputs [e.g., Petersen et al., 2013], oceanic eddies should be automatically recorded using these data and outputs [e.g., Yang et al., 2013; Sun et al., 2014; Pegliasco et al., 2015]. In general, the recording of oceanic eddies often includes two independent steps: automated eddy identification and automated eddy tracking. The eddies are identified in a sequence of SLA maps using an identification algorithm or identified from velocity fields. An automated tracking procedure is then applied to determine the trajectory of each eddy [Chelton et al., 2011b]. Several automated identification and tracking algorithms have been developed for eddies in the ocean  [Chelton et al., 2011b; Ienna et al., 2014; Mason et al., 2014; Yi et al., 2015].

For the eddy tracking stage, according to a recent census [Wang et al., 2015; Yi et al., 2015], approximately 10-30% of eddies may be found in proximity to a neighboring eddy in any given global SLA map and they frequently interact. Therefore, an eddy tracking process should have the capability to distinguish different dynamic processes (e.g., merging and splitting) during its dynamic evolution. Moreover, an eddy tracking process must be accurate and fast enough to handle a huge amount of data, which will be even larger in size if spatio-temporal resolution of observations and numerical simulations increases.

Implemented automated tracking procedures differ in detail, but they are all similar in concept because they utilize the nearest neighbor strategy [Chelton et al., 2011b]. For each eddy $E_i$ identified at time step $k$, the nearest eddy to $E_i$ at the next time step $k+1$ is identified as part of the trajectory of eddy $E_i$. A more advanced procedure uses eddy shape error as an additional condition when assessing an eddy trajectory [Mason et al., 2014].

However, there is a "missing eddy" problem that must be solved in the eddy tracking stage [Chelton et al., 2011b]. An eddy at time step $k$ may have no associated eddy at time step $k+1$, which is simply due to a temporary missing eddy in the identification process; this can occur for a variety of reasons related to sampling errors and measurement noise [Chelton et al., 2011b] or limitations of the eddy detection step when an eddy is too weak/small at a time step. Chelton and his colleagues made an attempt to accommodate such problems; they allowed for the reappearance of a temporarily missing eddy by looking ahead two or three time steps. Unfortunately, this "look-ahead" procedure considers too many nearby eddies as potential ones. In practice, the results of this simple "look-ahead" procedure were disappointing because the resulting eddy trajectories often jumped from one eddy track to another. As a result, the look-ahead approach was abandoned, even though it is a solution to the "missing eddy" problem [Chelton et al., 2011b].

Recently, the concept of Multiple Hypothesis Assignment (MHA) was introduced to solve the missing eddy problem by abandoning the simple closest eddy strategy and applying a new "look-ahead" procedure [Faghmous et al., 2013].

The MHA method can effectively solve the missing eddy problem in a straight-line model when the trajectory being
followed is a branch without any splitting, but it is algorithmically and computationally complex. Given the
maximum number of eddies in any time frame $M$, the number of look-ahead time steps $N$ (with $N=0$ being the
original linear closest eddy procedure without look-ahead) and the total number of time steps $T$, the MHA has a
larger time complexity (the total amount of time taken by an algorithm), $O$ ($M^{N+1}T$) at the worst-case [Faghmous et
al., 2013].
The existing straight-line model can trace the kinematic motion of eddy. The dynamic evolutionary processes (e.g.,
merging and splitting) of the eddy are, however, ignored by the model. This implies that each eddy $E_i$ identified at
time step $k$ has only one eddy as part of its trajectory at time step $k$-1 and has only one eddy as part of its trajectory
at time step $k$+1. In the ocean, small eddies may merge to form larger ones. As shown in Figure 1, the anticyclonic
eddies AC1 and AC2 observed on July 26, 2006 merged into a single one on July 31, 2006. Then, the cyclonic
eddies C1 and C2 on July 26, 2006 merged to form a larger one on August 3, 2006. To describe such processes, the
eddy tracking records should be trees with branches instead of simple straight lines.
To record the dynamic evolution of eddies, two fundamental algorithms are required. First, the two nearby eddies
should be distinguished in the identification stage using a segmentation strategy in which the target region is divided
into two corresponding eddies. Otherwise, the merging and splitting processes cannot be determined properly. This
problem was recently solved by the use of segmentation strategies, e.g., the close-distance segmentation strategy [Li
et al., 2014] and the watershed strategy [Li and Sun, 2015]. Because these segmentation strategies can distinguish
closed eddies, they can also potentially reduce the risk of having a missing eddy in the identification process.
Second, the merging and splitting processes in the tracking stage should be described in detail. We use a multi-
branch tree model to do so. The eddy $E_i$ identified at time step $k$ may arise from more than one eddy at time step $k$-1,
which subsequently merged; and $E_i$ may become more than one eddy at time step $k$+1 if it splits. We refer to this
model as the "Genealogical Evolution Model (GEM)" because it is a genealogical tree for recording the whole
evolutionary history of an eddy. The multi-way tree model in computer science can be used to store this type of
structure.
Moreover, the GEM also provides a new way to solve the missing eddy problem. Instead of the existing closest eddy
strategy, a temporal track tree with $N$ look-ahead time steps is used to maintain all possible tracks with the help of
the multi-way tree model. The method can effectively solve the missing eddy problem, regardless of whether the
eddy is splitting or not.
In this paper, we introduce the GEM to describe mesoscale eddies in a tracking process with a total number of time
steps $T$. The GEM allows the eddy to have multiple eddies as its parents or as its children in a multi-branch model. It
also solves the missing eddy problem by using a new look-ahead method similar to the MHA. Compared with the
time complexity $O$ ($M^{N+1}T$) of MHA, the new method is much faster and has much less time complexity $O$
($LM(N+1)T$), where $L$ denotes the pixels of target region. Besides, if the GEM was implemented with the computer
codes properly, the output data of GEM also record the dynamic evolution of the eddy in detail and will potentially
be useful for other research fields, e.g., the dynamics of cyclones in meteorology. As an example, The GEM is
applied to eddies in the North Pacific Ocean (NPO) only, and we assume the eddies do not cross the equator.
The paper is organized as follows. The data and eddy detection methods used in this study are introduced in section
2. Then GEM is introduced in section 3, including similarity vector, look-ahead approach and the worst-case
runtime complexity i.e. time complexity. Results including eddy tracks and examples of merging and splitting
events in a sample area in the North Pacific are shown in section 4. The impacts of data noise and parameters
uncertainties on the results are discussed in section 5. Finally, a summary and conclusions are given in section 6.

## 2   Eddy identification
### 2.1   Input data
The input data consists of the original altimetry field, which can come from satellite observations or numerical
simulations. The altimetry field used in this study is the 20-year (1993-2012) daily SLA data from the merged and
gridded satellite product of Maps of Sea Level Anomaly (MSLA) at $0.25^{\circ} \times 0.25^{\circ}$ resolution in the global ocean
from AVISO (http://www.aviso.oceanobs.com/). In this study, we use the "DT14" (delayed-time 2014) altimeter
product [Duacs/AVISO, 2014], which is adequate for direct eddy detection [Capet et al., 2014] though it still has
about 2-3 cm error globally for short temporal scales [Carrere et al., 2016]. A comprehensive discussion of gridded
Aviso products for eddy investigations can be found in Chelton et al. (2011b).
We used the original SLA data ("DT14") without any filtering or smoothing to identify eddies in this study.
However, this does not imply that data smoothing is not needed for the SLA data in related studies (e.g. eddy
detection, eddy tracking). For example, to calculate some eddy parameters (e.g., velocity and vorticity), smoothing
may be required, as pointed out by Chelton et al. (2011b). Moreover, the data errors, even if they are very small,
might affect the eddy detection (see discussion in section 5.1).
### 2.2   Eddy identification
The eddy identification used in this study is similar to those used before [Chelton et al., 2011b; Mason et al., 2014],
to identify eddies from SLA data. The eddies may be identified with multinuclear (two or more SLA extremes in
one eddy) or mononuclear (only one SLA extremum in one eddy).The following mononuclear eddy definition is also
similar to what was used by other authors [Chaigneau et al., 2011; Li et al., 2014; Li and Sun, 2015]. We have
adopted the eddy detection step from Li and Sun (2015), which provides us with the necessary input for the tracking
routines, namely eddy areas and boundaries. Each pixel has eight nearest neighbours. A point within the region is a
local extremum if it has an SLA greater or less than all of its nearest neighbours. We also use such definition of
extremum in our following analysis, in which the extrema are identified by checking each pixel in the map along
with the eight pixels around it. An eddy is defined as a simply-connected set of pixels that satisfies the following
criteria:
(1) The SLA value of all of the pixels is above (below) a given SLA threshold;
(2) Only *one* SLA extremum exists in the pixel set;
(3) The amplitude of the eddy (the max difference of SLA values) is larger than a critical value (e.g., 1 cm);
(4) The area of the eddy must be large enough for estimating eddy parameters (say >16 pixels).
Conditions (3)-(4) provide the lower bounds for eddy size and amplitude. These conditions automatically reduce the
total number of detected eddies. Condition (1) is the same as the first criterion in Chelton et al., (2011b). It is used in
consideration of the 2-3 cm of background SLA error [Carrere et al., 2016]; so, small fluctuations in SLA field
would not be taken as eddies in this study. Condition (3) was generally used previously [Chaigneau et al., 2011;
Chelton et al. 2011]. Condition (4) is more restrictive than the generally used value of eight pixels [e.g., Chelton et
al., 2011; Li et al., 2014]; so, this condition is an add-on, which is potentially useful when deriving eddy parameters
using a nonlinear optimal fitting method [Wang et al., 2015; Yi et al., 2015]. If the eddy area is too small (only a few
pixels), its parameters (e.g. amplitude, area, radius, etc.) are very sensitive to its area (number of pixels). Besides, we
don't put limits on eddy pixel number maximum (e.g., <1000) and eddy size (e.g., <400-1200 km) while such limits
were generally used previously [e.g., Chelton et al. 2011; Mason et al., 2014].
The SLA extremum so determined is called eddy center. The set of pixels belonging to an individual eddy is referred
to as the area of the eddy, and the outmost SLA contour is the boundary of the eddy. We use the area and boundary
to calculate the similarity of eddies in section 3.2.
Each eddy is identified by the following procedures. First, according to condition (1), we find a simply-connected
region with a given threshold of SLA >3 cm for cyclonic eddies and SLA<-3 cm for anticyclonic eddies. Second, we
check whether there is at least one extremum in the region. If the eddy is multinuclear, we use a segmentation
method to segment them to satisfy condition (2). Finally, we check whether the region satisfies the eddy conditions
(3) and (4); we remove those weak (amplitude < 1 cm) and small (pixels<16) eddies.

**2.3    Eddy segmentation for merging and splitting events**
Figure 2 illustrates the necessity for eddy segmentation based on the merging process of two eddies. Two different
mononuclear algorithms are used in the upper and lower rows. In the top panels of Figure 2, eddies are identified by
non-segmentation algorithm. Such mononuclear eddies may be very small. The time evolutions from t=1 to t=3
show a decay scenario of two closed eddies C1 and C2. Both their amplitudes and areas become smaller and smaller
with time. Then, a large eddy C3 suddenly appears in the same region without any premonition. It is hard to see
what happened during the time from t=1 to t=3 from the parameters (amplitude and area) of mononuclear eddies
identified by reducing the number of contours of the SLA until there is only one extreme in the contour (Chaigneau
et al., 2011) instead of the segmentation algorithm [Li and Sun, 2015]. In contrast, the bottom panels of Figure 2
show a merging scenario of two closed eddies C1 and C2 using the segmentation algorithm [Li and Sun, 2015].
During the time from t=1 to t=2, both their amplitudes and areas are only marginally changed, while their distance is
continually reduced. Then, a large eddy C3 naturally emerges in the same region, while C1 and C2 disappear. It is
recognized from the eddy data that C3 is the merging result of C1 and C2.
Figure 3 illustrates this eddy segmentation strategy. Figure 3a shows two individual but nearby eddies. The pixels
between the two dashed lines are naturally divided by the watershed (For basins, the "watershed" is a ridge between
them, while it is a valley for plateaus). As shown in Fig. 3b, the cross section of the eddy clearly shows that two
closely located pixels $P_1$ and $P_2$ on the left and right sides of watershed would slide along the path of steepest descent
in the map of SLA data to different eddy centres. The shape of SLA can provide sufficient information to segment
the multinuclear eddy into mononuclear ones.
Herein, we use the Mononuclear Eddy Identification (MEI) of the Universal Splitting Technology for Circulations
(USTC) with watershed segmentation [Li and Sun, 2015] and include in our code the calculation of eddy parameters,
including amplitude, radius, area, and boundary (Fig. 3), which might be potentially used in other studies [Sun et al.,
2014].
The output eddy parameters from MEI is then used as input for our novel tracking algorithm GEM. The GEM
mainly represents the logical relationship of eddies, which is less dependent on physical parameters which may
change greatly because of dynamic evolution (e.g., splitting, merging). To this end, the GEM takes the previously
identified eddies by MEI (with area/boundary, see section 2.2) as its input data.

**3    Dynamic tracking**
**3.1    Overview of GEM**
The GEM is a logical model used for tracking the dynamic evolution of mesoscale eddies in the ocean (Fig. 4). The
model essentially establishes logical relationships of previously identified eddies. The relationships are determined
by two relatively independent steps i.e. the GEM algorithm consists of two parts (see Fig.4 for details): first,
measuring the "map link" between two time steps and then connecting all time steps to the "track tree".
The first part of GEM is "map link" which uses as input eddy data obtained in the prior eddy identification step
(area/boundary, see section 2.2) to establish the link of an eddy from one temporal snapshot to the next, namely
living, missing, death, birth, and the associated dynamical processes of merging and splitting. In this part of the
work flow, we use a 2-D vector rather than a passive scalar to measure the similarity between eddies $E_1$ and $E_2$ on
two neighboring days (Figs. 5 and 6, see section 3.2.1 for details). We then use a relatively complex look-ahead
procedure to solve the missing eddy problem (section 3.2.2). This new look-ahead approach has a duration of $N$ days
(Fig. 7). Finally, the links of the eddies in different snapshots are saved (see section 3.2.2 for details).
The second part is "track tree", which uses the outputs from "map link" (i.e., eddy links), as its input (Fig. 4). It
connects the eddy links from branches to a tree with the genealogical model (Fig. 8) using two sub procedures:
"eddy branch" and "eddy tree". In the "eddy branch" part, we use *parent* and *child* to define the eddy relationship
and define all possible types of eddy states: birth, death, living, missing, merging and splitting (Fig. 8a).
Consequently, we identify different roles in the eddy branches (see section 3.3.1 for details). Finally, in the "eddy
tree" procedure, we connect the branches based on their roles (parent, child, and grandchild, etc.) in the genealogical
tree (Fig. 8b). The output of GEM includes eddy tracks and the records of eddy relationships (see section 3.3.2 for
details).
In short, the GEM uses previously identified eddies and/or their links to make dynamic tracks via a genealogical tree
model. In addition to eddy domain and boundary, it needs two parameters as input, the critical value of area ratio $r_c$
and $N$. See section 5.2 for discussion on the impacts of these parameter choices.

### 3.2 Map link

To establish the relationships between the previously identified eddies, the first part of GEM used evaluates the
similarity of these eddies which is defined here based on the overlap of the domain of an eddy in two consecutive
time steps. It begins with defining similarity based on the overlapping area of eddies in consecutive time steps.
Subsequently, the overlapping area which is closest to the one of the original eddy is defined to be the successor of
the original eddy (if the threshold is met).
3.2.1 Eddy similarity
At first, the eddy similarity is calculated with an example (Fig. 5a) before proceeding to the mathematical
expressions. There were three eddies A1, A2 and B1 detected on March 28, 1997. In Figure 5b, there were four
eddies, A1, A2, B1, and B2 on March 29, 1997. We overlapped the eddy domains into a single map (Fig. 5c). Then,
we used the intersection of eddy domains on different days to calculate the similarity. For eddies A1 and A2, the
intersections were very close to their respective domains on March 28 and 29. For eddy B1, the intersection was
close to the area on the second day, but it was only part of that on the first day. Consequently, eddies A1 and A2 had
full similarity on these days, while eddies B1 and B2 only had partial similarity on these days.
To estimate the above similarity, let us describe it in a mathematically logical way. As shown in Figure 6a, there is
an eddy ($E_1$) that is identified by the thick contour of Boundary 1 in the rectangular comparison region (not shown in
figure) on day 0, and there are three eddies ($E_2$, $E_3$ and $E_4$) that are identified in the same region on day 1. This
comparison region, which is centered at the eddy center of $E_1$, moves in time with the target eddy ($E_1$). To determine
the similarities between $E_1$ on day 0 and $E_2$ to $E_4$ on day 1, we intersect the domains of day 0 and day 1. For example,
to determine the similarity between $E_1$ and $E_2$, we count the overlap area $S_{12}$ (defined as the intersection of Boundary
1 and Boundary 2) between $E_1$ (area $S_1$) and $E_2$ (area $S_2$), and then we calculate the following ratios:
$$r_1 = S_{12} / S_1 , \qquad\qquad\qquad\qquad (1a)$$
$$r_2 = S_{12} / S_2 . \qquad\qquad\qquad\qquad (1b)$$
Clearly, the values of $r_1$ and $r_2$ are within [0, 1]. The larger $r_1$ and $r_2$ are, the larger possibility that $E_2$ has to be the
snapshot of $E_1$ on day 1. Eddy movement speeds are generally less than 0.1 m/s, which implies that an eddy can only
move one grid box ($0.25^o$) in 3-4 days. Thus, the overlap on different subsequent days of the same eddy area should
be large enough. This is one of the parameters to set. When we applied GEM to track eddies in the Northern Pacific
Ocean, we choose $r_c$=2/3, and the choice of $r_c$ is comprehensively addressed in section 5.2.
Using the vector ($r_1$, $r_2$) and the critical value $r_c$, we define four different types of similarity between two eddies (Fig.
6b). From low to high, they are as follows: Type 0 (T0: $r_1 < r_c$ and $r_2 < r_c$), where $E_1$ and E2 are unrelated; Type 1 (T1:
$r_1 > r_c$ and $r_2 < r_c$), where $E_1$ on day 0 is part of $E_2$ on day 1 ($E_1$ enlarging or merging); Type 2 (T2: $r_1 < r_c$ and $r_2 > r_c$),
where $E_2$ on day 1 is part of $E_1$ on day 0 ($E_1$ decaying or splitting); and Type 3 (T3: $r_1 > r_c$ and $r_2 > r_c$), where $E_1$ and
$E_2$ are the same eddy at different locations on different days ($E_1$ living and moving). The last type (T3, living) is
prescribed  in cases when the center of $E_1$ propagates less than a pixel toward that of $E_2$, because the eddy movement
speed is physically less than one grid ($0.25^o$) per day. For example, eddy B1 on March 29, 1997 in Figure 5b is
simply assigned to T3 (living) even though $r_1 < r_c$. Eventually, we obtain the relationships between $E_1$ and $E_3$ or $E_4$
(Fig. 6a). Because the present method uses a vector to express eddy similarity, we call it the similarity vector. This is
an alternative to scalar similarity parameters [e.g., Ienna et al., 2014; Mason et al., 2014].
For example, as shown in Figure 6a, the high similarity between $E_1$ and $E_2$ over a critical value $r_c$ (marked as T3
(living) in Fig. 6b) suggests an evolution from $E_1$ to $E_2$. This is similar for eddies $E_1$ and $E_3$, but with a different
splitting relationship (marked as T2 (splitting) in Fig. 6b). The relationship between eddies $E_1$ and $E_4$ is designated
as "unrelated" because of the overlap in their areas is small or zero. In other words, their overlap rates are below the
critical value $r_c$ (marked as T0 in Fig. 6).
In previous eddy tracking studies, simple methods were used for weekly SLA data (delayed-time 2010), e.g., the
closest distance between eddies [Chelton et al., 2011b; Yi et al., 2015], the closest angle between eddies [Zhang et
al., 2014] and the dimensionless similarity scalar [Chaigneau et al., 2008; Mason et al., 2014]. There is always a risk
of eddy jumping (from one track to another) in these methods, except for that of Pegliasco et al. (2015), who used
intersections of eddy boundaries to find the continuing eddy. Compared to the previous tracking methods, we use a
more robust technique to assess the relationship of eddies in subsequent time steps by using the overlap of their
areas. In addition, we do not simply assign the continuing eddy using the similarity vector for the two adjacent days;
rather, we try to solve the temporary missing eddy problem by looking ahead a few days.
3.2.2     Eddy Look-ahead
In contrast to the procedure used in Chelton et al. (2011b), we use a relatively complex look-ahead procedure. An
example for a given eddy are shown in Figure 7a. In the upper row, both Ec1 and Ec2 take the same eddy Ec3 as
their subsequent T1 type of eddy, which is a merging event (e.g., eddies C1 and C2 in Fig. 1). Since a T1 (merging)
eddy has $r_2 < r_c$ (intersection only takes a part of the eddy Ec3 on day 1), two or more eddies (e.g., Ec1 and Ec2) on
day 0 could identify the same eddy (Ec3) as T1 eddy simultaneously on day 1. In the middle row, eddy E1 has two
T2 (splitting) type of eddies (Ec2, Ec3) at the same time; this is a splitting event (e.g., eddies B1 and B2 in Fig. 5).
In the lower row, eddy E1 has T2 (splitting) and T3 (living) types of eddies (respectively Ec2, Ec3) at the same time.
Although there may be many possibilities for any given eddy, there is at most one eddy that can be marked as a T1
(merging) or T3 (living) eddy on the following day (as $r_l > r_c = 2/3$ holds).
This new look-ahead approach with $N=2$ is shown in Figure 7b. After finishing the calculation of the following
eddies on day 1, we continue to calculate eddies on the following days. At this preparation stage, it is similar to the
MHA method with important modifications [Faghmous et al., 2013]. What makes this look-ahead procedure novel
and efficient is that we use two simple rules to directly choose only one day's result for the following eddies. Thus,
the procedure becomes linear without iteration, and it is much faster than the MHA, as discussed in the subsection
on the time complexity (section 3.4).
The two selection rules are: 1) the most similar, and 2) the earliest day. Rule 1 has priority. We first choose the most
similar eddy as the potential successor of Ed1 according to their types. According to Figure 6b, T2 (splitting) type
eddy covers only part of the original eddy while T1 (merging) eddy covers most part of the original eddy. The
similarity from low to high is T2<T1<T3. For example, if there is only one T3 (living) eddy in these days, we
choose it as the potential one. However, if there is more than one day with the same type of eddies, we need an
additional rule: the earliest day. For example, in the upper row of Figure 7b, there is one T3 (living) eddy on day 1
and one T3 (living) eddy on day 2, and there are two T2 (splitting) eddies on day 3. In this case, we choose day 1 as
the following day and the T3 (living) eddy as the following Ed1. In the middle and the lower rows, we choose day 2
and day 3 as the following days and the corresponding T3 (living) eddies as the following Ed2, Ed3 respectively.

## 3.3     Track tree

### 3.3.1     Eddy branch

After having determined the next subsequent days and the relationship types between eddies based on the above
process, we can now establish the branches of an eddy from one day to the next. Eddy branch describes the
relationship between two eddies at two different time steps. To describe the GEM more precisely, we use *parent* and
*child* to identify the different roles that the eddy plays in eddy branches. There are three types of logical
relationships used in GEM, as shown in Figure 8a.
The upper row shows a successor relationship: an eddy P on day 1 has only one successor (eddy P itself) on day 2.
In this case, eddy P is allowed to be missing during day 1 and day 2. Additionally, eddy P will be recorded as death
(black circle), if no successor eddy is found after $N$ days.
In the middle row, two (or more) eddies merge into one. The first type includes principal and subordinate merging.
A principal eddy $P_1$ and a subordinate eddy $P_2$ on day 1 merge into a larger eddy $P_1$ on day 2, whereas $P_2$ is recorded
as death. This occurs when a large eddy meets and merges with a small eddy (e.g., C1 and C2 in Fig. 1). The
anticyclonic eddies A1 and A2 in Fig. 11 also experience a similar process (see section 4.2 for details). The second
type is coordinated merging. Two (or more) parent eddies $P_1$ and $P_2$ merge to produce a new child eddy C, and all of
the parent eddies are recorded as death. This is because the similarity is not sufficiently high for either eddy to which
the record of eddy C should be appended. There might be another choice by keeping parent eddies $P_1$ and $P_2$ alive
and appending the record of eddy C to both eddies. This choice artificially increases lifetimes of eddy P1 and P2and
leads to other tracking problems; so, we abandon it.
In the lower row, a parent eddy splits into several child eddies. The first type is principal and subordinate splitting. A
parent eddy P splits into a relatively large eddy P (itself, i.e. the similarity type is T3 between the two eddies) and a
relatively small child eddy C (i.e. the similarity type between parent eddy P and child eddy C is a splitting
relationship T2), which is recorded as birth. The second type is coordinated splitting. Two (or more) child eddies are
born from the parent eddy P, which is then recorded as death. This occurs when all the similarity types between
child eddies and parent eddy are type 2 (T2).
3.3.2    Eddy tree
Finally, the track tree is recorded by connecting the eddy branches (Fig. 8b). Track tree of an eddy records
information of all the associated eddies (e.g., living, death, birth, merging and splitting, etc.) during its entire life. In
this process, the role that an eddy plays in the track tree is considered. The first generation is the parent eddy (e.g.,
$P_1$), the second generation is the child eddy (e.g., $C_1$) and the third generation is the grandchild eddy (e.g., $G_1$). The
track tree uses the above eddy branches (Fig. 8a). We connect the branches from one time to another to obtain the
whole eddy track tree.
There are two additional notations. First, an eddy emerging from the same family of eddies (e.g., two siblings $C_2$ and
$C_4$) will be recorded as a new family member (e.g., eddy $C_5$). Second, an eddy merging from two different families
of eddies (e.g., $C_1$ and $P_2$) will be recorded as a new eddy $N_1$.
Although the model could have several generations, we only recorded two generations i.e. parent and child in this
study due to the complexity of the output data structure and the time complexity. However, we can indirectly track
other generations using the relationships between them.

## 3.4    Time complexity

To calculate similarity vectors, we need to overlap two small regions around eddy E1. The total number of pixels in
the rectangular comparison region is $L$. The time complexity of the similarity vector is $O(L)$ for each day. If we use
$N$ look-ahead time steps to find the best choice, the time complexity of the branches will be $O(L(N+1))$ for one
eddy. Because all of the steps are linear without iteration, given the maximum number of eddies in any time frame
$M$, the number of look-ahead time steps $N$ and the total number of time steps $T$, the total time complexity is $O$
$(LM(N+1)T)$. The GEM algorithm can hardly be made any faster. When the number of look-ahead time steps $N$ is
more than one, the time complexity is much faster than $O(M^{N+1}T)$ of MHA.
For example, both $L$ and $M$ are approximately 1000, and $N$=2 is used in the present study. The MHA method will
require on the order of $10^2$-$10^3$ times more computational time than the present method; and the larger the value of $N$,
the more efficient the present method is. The look-ahead time $N$ may be potentially as large as one week ($N$=6), as
noted in the following discussion. Thus, the present method is especially effective compared to the previously
suggested methods when a long look-ahead time is required for poorly identified eddies.

**4 Results**
**4.1 Eddy tracks**
We first apply the MEI to detect the ocean eddies in the North Pacific Ocean (NPO) during 1993-2012. The eddy
centers (SLA extrema of eddy snapshots) on each day are counted on each $1^o \times 1^o$ grid. In general, anticyclonic eddies
are significantly more frequent than cyclonic eddies. As shown in Figure 9a, the cyclonic eddies are mainly located
in the western part of the NPO. For example, there are lots of cyclonic eddies east of Japan near the Kuroshio, which
can also be seen from both Figure 1 and the results in section 5.1. In contrast, anticyclonic eddies are mainly located
in the eastern part of the NPO (Fig. 9b). For example, the eddies are mainly anticyclones in the red box, which can
also be seen from the results in section 4.2. In general, the eddies are ubiquitous in Figure 9c (about 50-70 eddies per
year on each $1^o \times 1^o$ grid), except that there are several regions where both types of eddies are relatively scarce. One
of them is known as "eddy desert" (black box in Figure 9c) [Chelton et al., 2007]. The other region is the North
Equatorial Current (NEC) (blue box in Figure 9c) [Hu et al., 2015]. Finally, we present in Figure 9d the ratio of
difference of the numbers of cyclonic and anticyclonic eddies to the total number of eddies.
We apply the GEM to these eddies detected by MEI with $r_c$=2/3 and $N$=2. In the NPO, there are a total of 60276
eddies with lifetimes longer than 30 days. Among them, 37553 of the eddies are anticyclonic and 22723 are cyclonic.
The tracks of long-lived eddies are plotted in Figure 10. In general, they are similar to those shown in previous
studies [Chelton et al., 2011b]. There are 7290 anticyclonic and 3627 cyclonic eddies with lifetimes longer than 100
days (Fig. 10a), and the ratio of anticyclonic to cyclonic eddies is approximately 2. The ratio is larger for eddy
lifetimes greater than 400 days, which was also noted in previous studies [Chelton et al., 2011b; Xu et al., 2011].
Each track is very smooth because we require that the snapshots of eddies on different days overlap one another. We
have done a visual evaluation of many long-lifetime eddy trajectories and the quality of the tracking results is
reasonable. We will take the long-lived C1 in Figure 10b as an example.
Eddy C1 was first detected as an eddy initiated on September 14, 1995, with an extremum at 163.5$^o$W, 10.5$^o$N. It
then travelled to the northwest and disappeared at 151.25$^o$W, 20.5$^o$N on March 11, 1997. Its trajectory is the longest
that we have detected in the NPO (Fig. 10b). The trajectory is smooth, except for a sudden jump from 167.5$^o$E to
166.75$^o$E (Fig. 10c) on July 31, 1996. The GEM algorithm did very well at whether we should connect the
trajectories from before July 30, 1996 with that after July 31, 1996, into a single trajectory.
To clarify this, we plot the two SLA fields in Figure 10d. The SLA field on July 30, 1996 is plotted as contours. The
eddy center is marked by a black cross at 167.5°E, 16.5°N. In contrast, the SLA field on July 31, 1996 is plotted in
shading. The eddy center is marked by a red cross at 166.75°E, 17.25°N. The distance between the eddy extrema was
larger than 100 km within a day. Although that distance is far beyond the criterion applied in standard eddy tracking
routines [Mason et al., 2014; Yi et al., 2015], we can see from the SLA fields that they both indicated the same eddy,
and that it was consistent with our approach to connect the trajectories into a single trajectory.
There may be no associated eddy can be identified at the next time step for an eddy at time step $k$, and it may be the
result of eddies temporarily "disappearing" for a variety of reasons related to sampling errors and measurement
noise [Chelton et al., 2011b]. The application of similarity vector and look-ahead procedure can effectively
accommodate such problems and allow for the reappearance of temporarily "disappearing" eddy in the tracking
procedure. In turn, the application of similarity vector reduces the usage of the look-ahead procedure. It is clear that
the similarity expressed as a vector is better than that as scalar using simple distance.

## 4.2    Eddy merging and splitting

The trajectories provide evidence of dynamic evolution. The time evolution of a couple of anticyclonic eddies is
depicted in Figure 11a, which implies a merging process occurring in the red boxes in Fig. 9. As shown in Figure
11a, eddy A1 had a westward movement with a speed of 2.6 cm/s, and eddy A2 lingered near 133°W. Then, both
eddies merged into one large eddy on April 23, 1997. That evolutionary process is clearly shown by the SLA fields
(Figs. 11c-j). In Figure 11c, there were two anticyclonic eddies, A1 and A2, located at 132°W, 28.5°N. Eddy A1
moved from east to west with a nearly constant speed of 2.6 cm/s, whereas eddy A2 had negligible zonal motion.
They then rotated clockwise about each other with an average angular velocity of $6\times10^{-7}$ $s^{-1}$, as denoted by the blue
arrows. Finally, they merged into the new large eddy A2 (see animation in supplement).
The SLA field shows that an eddy splitting process also occurred in the box the same time. The time evolutions of
anticyclonic eddies B1, B2 and B3 are depicted in Figure 11b. At first, eddy B1 had a fast westward speed of 10.4
cm/s. It then split into two eddies (B1 and B2) on March 29, 1997 (Fig. 6). Eddy B1 traveled at its original speed
whereas eddy B2 lingered at its origin. Then, eddy B3 emerged at a location between B1 and B2 on April 9, 1997,
which slowed down the speed of B1 to approximately 3.5 cm/s. After that, eddies B2 and B3 merged into a new
eddy B3 on April 19, 1997. In fact, similar to eddies A1 and A2, eddies B1 and B2 eventually merged into a new
eddy on May 20, 1997 (not shown). The SLA maps in Figures 11c-j show more details that were not recorded by the
eddy tracking data. Note that eddy B2 had a very short lifetime of 20 days but a complex dynamic process. If only
long-term eddies (lifetime > 30 days) were saved, the corresponding evolutionary process might not be recorded
properly.
It is expected that a pair of cyclonic eddies will have a counter-clockwise rotation in the Northern Hemisphere,
which is known as the Fujiwhara effect for atmospheric cyclones [Fujiwhara, 1921]. When two cyclones are close
enough, they will begin to orbit cyclonically (counter-clockwise in the Northern Hemisphere). Because the above-
mentioned eddies are anticyclonic, they have opposing directions of rotation, which appear as two point vortices
moving in circular paths about the center of vorticity in classical fluid dynamics [Batchelor, 1967].
### 4.3 Census of merging and splitting events
To illuminate how often the merging and splitting processes occurred, we counted the total number of merging and
splitting events on each $1°×1°$ grid each year. The merging and splitting events were homogeneously distributed in
the oceans, but in general were very few times each year per $1°×1°$ grid element. The merging frequencies for
cyclonic eddies and anticyclonic eddies are shown in Figure 12, which are similar to their splitting frequencies (not
shown). The distribution pattern of merging frequencies for cyclonic eddies in Figure 12a, is very similar to that of
cyclonic eddy centers in Figure 9a. In contrast, the merging frequency for anticyclonic eddies was larger along the
west coast (Fig. 12b), whereas the anticyclonic eddy centers were located mainly in the east (Fig. 9b). Although
merging and splitting events may occur anywhere in the ocean there is spatial variation in the number of events (Fig
12c, d)
The first type of special region is the western boundary. It is known that the western boundary is a sink of eddy
energy caused by the interaction with the bottom and lateral topography [Zhai et al., 2010]. It is also known as a
"graveyard" for westward-propagating ocean eddies [Zhai et al., 2010; Chelton et al., 2011b]. The second type of
special region is located in strong currents, including the Kuroshio Current, and the NEC [Hu et al., 2015]. Among
those currents, the eddies in the NEC had the highest frequency of merging and splitting events, which was not
noted in previous studies. The third type of special region is located in the northeast Pacific, which is also known as
an "eddy desert" [Chelton et al., 2007]. The fourth type of special region is located in enclosed marginal seas,
especially the Bering Sea.
By comparing Figure 12 with Figure 10, we can see that the regions with high frequencies of merging and splitting
events have relatively few eddy tracks, especially in the NEC (blue box in Figure 9c) and in the "eddy desert" (black
box in Figure 9c) in the northeast Pacific. The existence of "eddy desert" may be due to the fact that the eddy was
too small to be detected or the fact that the eddy lifetime was too short [Chelton et al., 2011b]. However, Figures 9
and 12 suggest that merging and splitting events may be a major contributor to the "eddy desert".
We also calculate the average dynamic (merging and splitting) events per eddy as a function of lifetime (Figure 13).
The results are similar regardless eddy polarizations and dynamic types. The merging and splitting events are
approximately linear increase with eddy lifetime. However, the anticyclonic eddies seem more vigorous in ocean
dynamics than cyclonic eddies.

## 5    Discussion

### 5.1    Data noise

Although "the Aviso product DT14" is much better than previous products, there are still some notable errors, especially for short temporal scales of less than two months [Carrere et al., 2016]. It was reported that there are along-track SLA errors of about 2-3 cm globally and of more than 3 cm at high latitudes and in shallow waters.

To reduce the noise in SLA data, one may use the Gaussian structure filter [Chelton et al., 2011b; Mason et al., 2014], Hanning filters [Penven et al., 2005], or Lanczos filter [Chaigneau et al., 2008]. As certain parameters need to be chosen in these filters, the filtered results depend much on these parameters [see Fig. A1 in Chelton et al., 2011b]. As sensitivity test we apply a simple five-point quadratic smoothing to the SLA data. The filtered data are then piecewise $C^2$-smoothed by a quadratic function, which satisfies the potential requirements for calculating vorticity (second derivative of SLA) from SLA data.

Figure 14 shows the non-smoothed and smoothed SLA data from January 1, 1993 to January 4, 1993. The smoothed SLA maps are very close to the non-smoothed SLA maps. And the values at the SLA extrema (not shown) are close to their original values. This implies that the noise in the DT14 data is sufficiently small for our purpose.

However, the noise cannot be neglected, even when they are small. They might induce additional SLA extrema (see the definition of extremum in section 2.2), which eventually affect eddy detection, e.g., the additional extremum on January 2, 1993 in box A and the additional extremum on January 3, 1993 in box B (Figure 14). These additional extrema existed only for a very short period (one or two days). But they can induce additional merging and splitting events, which may cause eddies to unexpectedly terminate [Chelton et al., 2011b]. The ambiguity of the eddy identification procedure, which may be caused by sampling errors and measurement noise in the input SLA data, strongly suggest the application of a look-ahead approach.

### 5.2    Impact of variations of parameters

To discuss the impact of the GEM critical value $r_c$ and look-ahead time $N$, we carry out a sensitivity study in the north Pacific from year 1993 to 2012. The number of eddies with lifetimes > 30 days is counted for different $r_c$ and $N$, as shown in Figure 15a. Note that the results are very similar, except for $N=0$ (i.e., without any look-ahead). It is from the above discussion that we see look-ahead is necessary when there are extrema due to small noise in the data. The number of eddies does not change substantially with $r_c$ for any $N>1$, when $r_c$ is within 0.5 to 0.8 (e.g. 63469 eddies were identified with $N=2$, $r_c$ =0.5 and the identified eddies number was 63630 with $N=4$, $r_c=0.8$.).Meanwhile, the numbers of merging and splitting events are also counted for different $r_c$ and $N$, as shown in Figure 15b. In general, the splitting events occurs slightly more frequently than the merging events (e.g. 151220 splitting events and 150612 merging events for $N=2$, $r_c$ =0.5). Note also that the results are very similar, except for $N=0$. The numbers of merging and splitting events seem to converge for $r_c$ >0.5 as $N$ increases. For each $N>0$, the numbers of merging and splitting events reach a maximum at $r_c$ =0.6. A relatively loose similarity condition ($r_c$ <0.5) will lead to

a risk of eddy jumping from one track to another, which consequently reduces both total eddy number and dynamic
events. On the other hand, a relatively strict similarity condition ($r_c > 0.9$) will lead to a risk of missing eddies, which
may also reduce both total eddy numbers and dynamic events.
In general, one would like the tracking results to be insensitive to the choice of these parameters. From Figure 15,
we can observe that $0.5 < r_c < 0.8$ appears to be a choice with relatively robust results. The optimal value for $r_c$ might
be 0.6-0.7, which is reasonable. In one hand, we first require that $r_c > 0.5$. On the other hand, we know there is area
error in calculation (~10%) since only eddy grids are taken into consider. This is also the reason why we need $r_c$
<0.9 or even smaller. So the optimal value should be within 0.5-0.9, and ~0.7 is just in this middle. We also find that
the look-ahead time $N$ should be larger than 0; otherwise, the risks of eddy jumping and eddy missing are too great.
The look-ahead approach effectively reduces such risks. For example, $N=1$ and $N=2$ have 95.5% and 98% of the
total eddies for $N=4$, respectively. To reduce the missing eddies to 1%, the look-ahead time might be greater than six
days. This is also the physical requirement of the representative period of the merged SLA data [Chelton et al.,
2011b]. Although $N=4$ might be better, $N=2$ produced a very similar result (~2% bias to $N=4$) and with a
significantly lower computational cost. Our present parameters are reasonable considering of computational cost.
It should be pointed out that GEM is relatively independent to MEI, but the ratios $r_1$, $r_2$ and $r_c$ might be sensitive to
the method used in identification. We noted that GEM based on Okubo–Weiss (O–W) parameter identification is
much sensitive to the critical value $r_c$ than SLA based one, since O-W based eddies are much smaller and more
possible to be unreal [Chaigneau et al., 2008]. Besides, $r_c$ may not be independent with $N$, and the present $r_c$ should
only be valid for small time steps. If the time step is too large, the distance of eddy motion may be too far. And eddy
snapshots can't overlap with each other. This constrain for time step is something like the Courant–Friedrichs–Lewy
(CFL) condition (for time step) in computer fluid dynamics. In general, we think any tracking method should have
this time-step limitation (depending on eddy size/propagation speed), if one don't want to mix one signal with
another.
Finally, as noted in section 4.2, there are short-term eddies (lifetime < 30 days), which might experience complex
evolution process. If only long-term eddies (lifetime > 30 days) were saved, the corresponding evolution process
might not be recorded properly. This should be noted in further applications on eddy dynamics with satellite
altimetry data.
**5.3   Impact of eddy boundary**
It is difficult to directly compare the influences of eddy boundary due to parameter choice in eddy identification. We
can, however, estimate the influence of the eddy boundary using an indirect way. Because the eddy center is
relatively robust, different identification methods mainly give different eddy boundaries. Consequently, the eddy
area $S$ is most sensitive to such an eddy area. However, the area ratio reduces the sensitivity to the eddy area $S$
because both the overlap area $S_{12}$ and the eddy area $S$ change synchronously. Moreover, our tracking results
fortunately are not very sensitive to $r_c$ (or the eddy area $S$), as noted in the above discussion. For example, the
present results are based on a very strict identification method. If we modify the threshold of eddy amplitude from 1
cm to 3 cm, the number of identified eddies will decline. Nevertheless, the identification results for the long-lived
eddies appear to be similar (Table 1).
However, such sensitive test may be only valid for the comparison of different parameter values in a same
identification method. It can't be simply extended to the comparison of eddies identified by different methods, since
the eddy detection algorithms differ a lot from each other. In general, the automated eddy detection algorithms are
categorized into three types: 1) physical parameter-based algorithms, e.g., Okubo–Weiss (O–W) parameter [Isern-
Fontanet et al., 2003; Chaigneau et al., 2008]; 2) flow geometry-based algorithms [Chaigneau et al., 2011; Chelton
et al., 2011b; Wang et al., 2015]; and 3) hybrid methods, which involve physical parameters and flow geometry
characteristics [Nencioli et al., 2010; Xiu et al., 2010; Dong et al., 2011; Yi et al., 2015]. For example, Yi et al.
(2015) used the O–W parameter to identify eddy kernels and SLA contour geometries to identify eddy boundaries.
So it is difficult to compare the influences of eddy territory by using different identification and tracking algorithms.

## 5.4    Future research

The GEM is a flexible model that can easily work with other relevant programs, e.g., data filtering and smoothing
algorithms [Chelton et al., 2011b; Ienna et al., 2014; Wang et al., 2014], other hybrid eddy detection algorithms [e.g.,
Yi et al., 2015] and O-W parameter detection [e.g., Petersen et al., 2013], because the GEM only requires a flow
field and previously identified eddies to accomplish dynamic tracking. In addition, the similarity measurement can
be replaced by similar methods [e.g., Pegliasco et al., 2015] when considering more complex conditions.
The identified eddies by using other identification algorithm without watershed can also be tracked with the GEM.
In this case, the strong interaction stage of eddies "in conjunction", which leads to genesis and termination of eddies,
is more likely missed as pointed out in section 2.3. However, the weak interaction stage of eddies (watershed free) in
some far distance could still be recorded, because most of merging/splitting records occurred at the interaction of
two eddies with a certain distance. This weak interaction still can't be recorded by previously interaction-free
tracking algorithm, which records only the isolated tracks. Thus the GEM extends the potential applications of
previously identified eddies.
The GEM is a complex model. The output data include eddy tracks, relationships and previously identified eddy
characteristics (e.g., amplitude and radius). These eddy characteristics, which were directly obtained from the
identification process, are useful for censuses [Chelton et al., 2011b]. However, they may not be sufficiently
accurate for some applications. For example, eddy area was required in our recent studies on typhoons and oceanic
eddy interactions [Sun et al., 2010, 2012, 2014]. Besides, some physical quantity (circulation, angular momentum,
energy) are required to be accurately calculated in the investigation of eddy dynamics process. A better way to
obtain these characteristics might be to use a nonlinear fitting of the flow field [Wang et al., 2015; Yi et al., 2015]
with appropriate models [e.g., Sun, 2011; Zhang et al., 2013] other than simply estimated from identification.
Another future research direction may involve comparing different tracking datasets. Because there are several
tracking datasets produced by various methods, it is useful to inter-compare them. This may improve both the
tracking methods and the available datasets for further studies.
The GEM can be easily applied to larger datasets, even to 3-D numerical simulation outputs [Petersen et al., 2013;
Woodring et al., 2016], because its computational time increases only linearly as a function of the size of the dataset.
The computation of the 20-year daily global SLA data only required a few hours on a personal computer. In a
personal computer with CPU of i7-6700k and 4.00 GHz, it takes about 15 minutes to identify snapshots of eddies,
about 20 minutes to establish similarity, and about 10 minutes to track eddies in the North Pacific Ocean (NPO) with
$0.25^o \times 0.25^o$ resolution of 20-year daily "DT14" data. Such a model can be used to analyze numerical simulation
outputs.
The GEM opens a window to investigate eddy dynamics [Wang et al., 2015] and other applications [Sun et al., 2014]
on those problems, e.g. (i) the strong eddy interaction which leads to genesis and termination of eddies (ii) the weak
eddy interaction which associates with merging/splitting events (iii) the weak eddy interaction which modulates the
eddy track and motion. As illuminated in Figure 11, the dynamic evolution of eddies is accompanied by abundant
phenomena that might be identified using the GEM. The present study is only the beginning of such applications.

**6    Conclusions**
We have introduced the GEM for the tracking of the dynamic evolution of mesoscale eddies in the ocean. Several
novel approaches (e.g., vector similarity and look-ahead approach) were applied to deal with unsolved problems in
tracking. All of the computational steps in GEM are linear and do not require iteration. Given the grid number of the
target region $L$, the maximum number of eddies $M$, the number of look-ahead time steps $N$, and the total time steps $T$,
the total time complexity is of $O$ $(LM(N+1)T)$. We applied the GEM to the eddies in the north Pacific. Eddy tracks
were smooth because we required that the snapshots of eddies on neighboring days overlap one another. Both
merging and splitting rates of eddies were high, especially at the western boundary, in strong currents and in "eddy
deserts". The GEM is useful not only for satellite-based observational data but also for the output of numerical
simulations. It potentially has many applications for studies of dynamic processes in related fields, e.g., the
dynamics of cyclones in meteorology. The "MEI" and "GEM" computer codes and program manual will be
provided openly at the website https://www.researchgate.net/profile/Liang_Sun20/ after publication of this
paper.


**Acknowledgements**
We thank the anonymous referees and Dr. John M. Huthnance for their comments and suggestions. We thank the
AVISO for providing the SLA data (http://www.aviso.oceanobs.com/). This work was supported by the National
Basic Research Program of China (Nos. 2012CB417402 and 2013CB430303), the National Foundation of Natural
Science (No. 41376017) and the Open Fund of the State Key Laboratory of Satellite Ocean Environment Dynamics
(No. SOED1501).

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

 Table 1. The census of long-lived eddies, where "Amp" represents the amplitude threshold used in eddy detection;
and "C" and "AC," respectively, represent cyclonic and anticyclonic eddies.

| Amp | AC (>100 d) | C (>100 d) | AC (>400 d) | C (>400 d) |
|------|------------|-----------|------------|-----------|
| 1 cm | 7290 | 3627 | 198 | 22 |
| 3 cm | 7118 | 3550 | 194 | 21 |


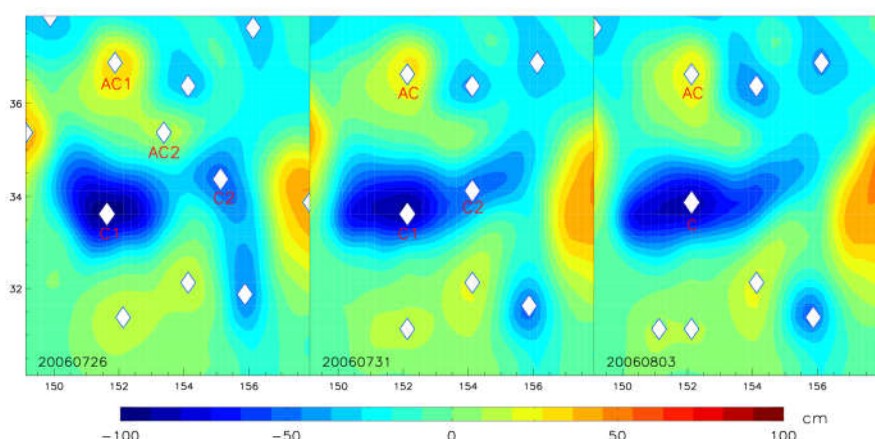


Figure 1. The evolutions of amplitudes and areas of eddies from July 5 to August 3, 2006 (after Li et al. 2014),
where the background field shows SLA, and white dots mark eddy centers. Two anticyclonic eddies AC1 and
AC2 merged into a single eddy on July 31, 2006. And, two cyclonic eddies C1 and C2 merged into a single one on
August 3, 2006.

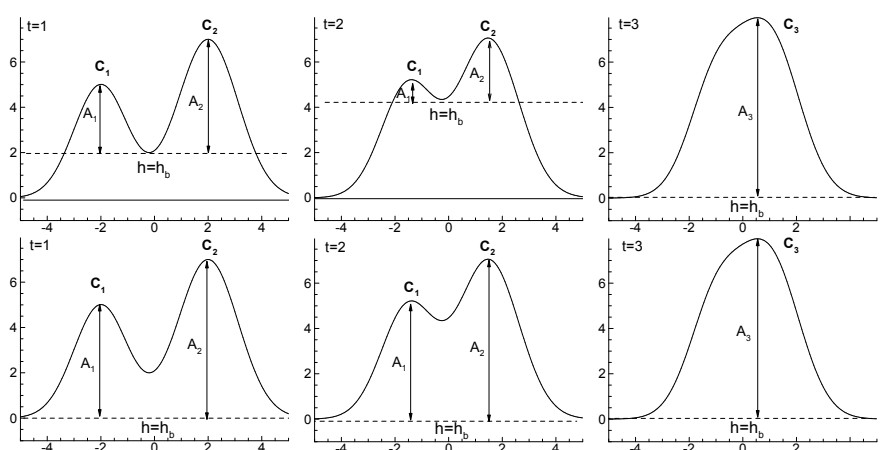


Figure 2. Top panels: Time evolution of two merging eddies revealed by the mononuclear eddy identification
without segmentation. Bottom panels: Time evolution of two merging eddies revealed by the mononuclear eddy
identification with segmentation. The h represents background SLA value, A represents amplitude of eddy,
and t represents the map at different time.

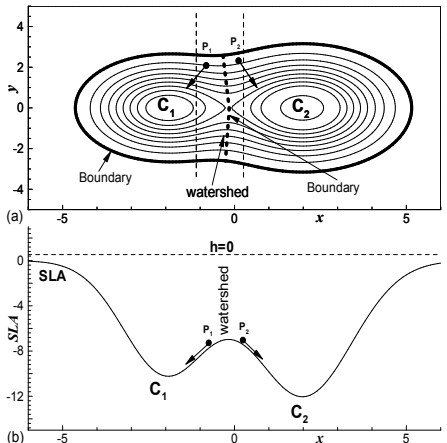


Figure 3. (a) Watershed as the natural division of eddies $C_1$ and $C_2$ from top view, where contours represent SLA. (b)
The particles $P_1$ and $P_2$ on the watershed flow downward to the eddy centres $C_1$ and $C_2$ from cross-section view.
After Li and Sun (2015).

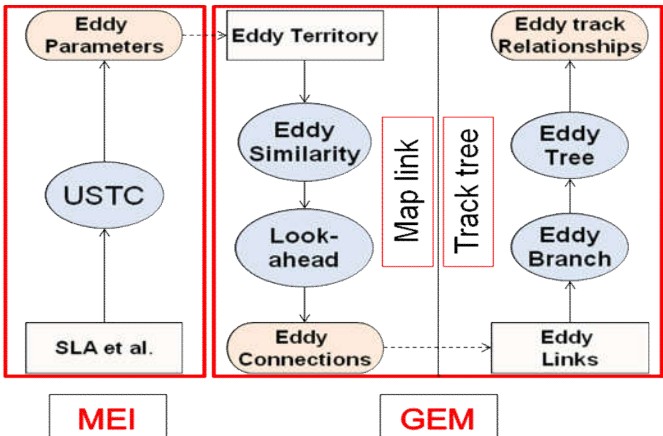


Figure 4. Flow chart of the systems. Mononuclear Eddy Identification (MEI) uses SLA to identify eddies via the
Universal Splitting Technology for Circulations (USTC) method. The GEM, which has two independent parts of
"Map link" and "Track tree", then uses the previously identified eddies for tracking.



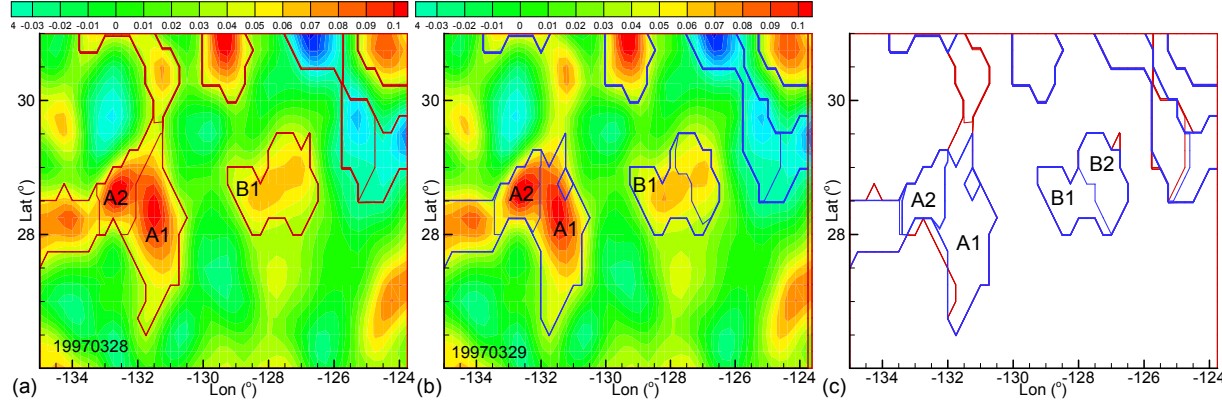

Figure 5. Sketch of eddy overlaps. (a) The SLA map (shading) and the boundary of eddies (red curves) on March 28, 1997, where A1, A2 and B1 represent identified eddies. (b)The SLA map (shading) and the boundary of eddies (blue curves) on March 29, 1997. (c) The intersection of eddy areas by overlap eddy identification maps.

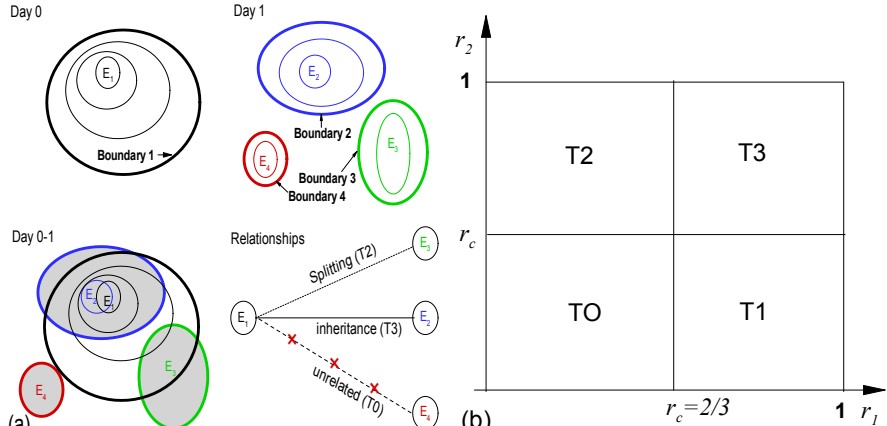

Figure 6. Sketch of eddy similarities. (a) The sketch of eddy overlaps. Eddy $E_1$ (black) is the eddy identified on day 0, where the thin contours represent the eddy parameter (e.g., the SLA value). The thick contour represents the eddy boundary. Eddies $E_2$ (blue), $E_3$ (green) and $E_4$ (red) are identified on day 1. We consider the overlay between the two eddies on different days to evaluate the similarity between them. (b) There are four similarity types (T0-T3) according to the values of $r_1$, $r_2$ and the critical value $r_c$, there is at most one eddy that can be marked as a T1 (merging) or T3 (living) eddy on the following day.

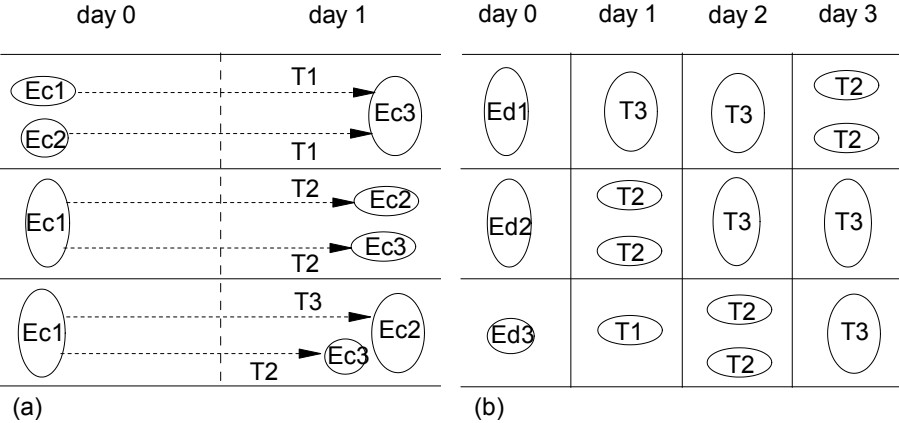

699

Figure 7. (a) Three typical cases of successors (T1, T2 and T3) from one day (day 0) to another (day 1). (b) The eddy at day 0 may have different successors corresponding to different numbers of "look-ahead" days, e.g., Ed1 at day 0 may have a T3 eddy on day 2, and have two T2 eddies on day 3.

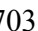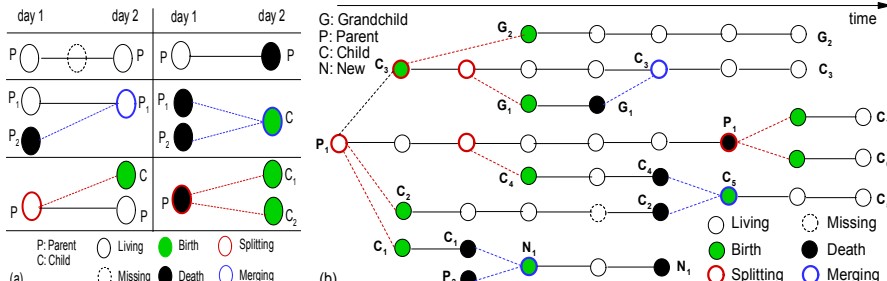

704

Figure 8. The logical genealogy of an ocean eddy with six states: birth, death, living, missing, splitting, and merging. (a) The logical relationships of eddies between two days. (b) The logical genealogy evolution model of an example eddy.

709

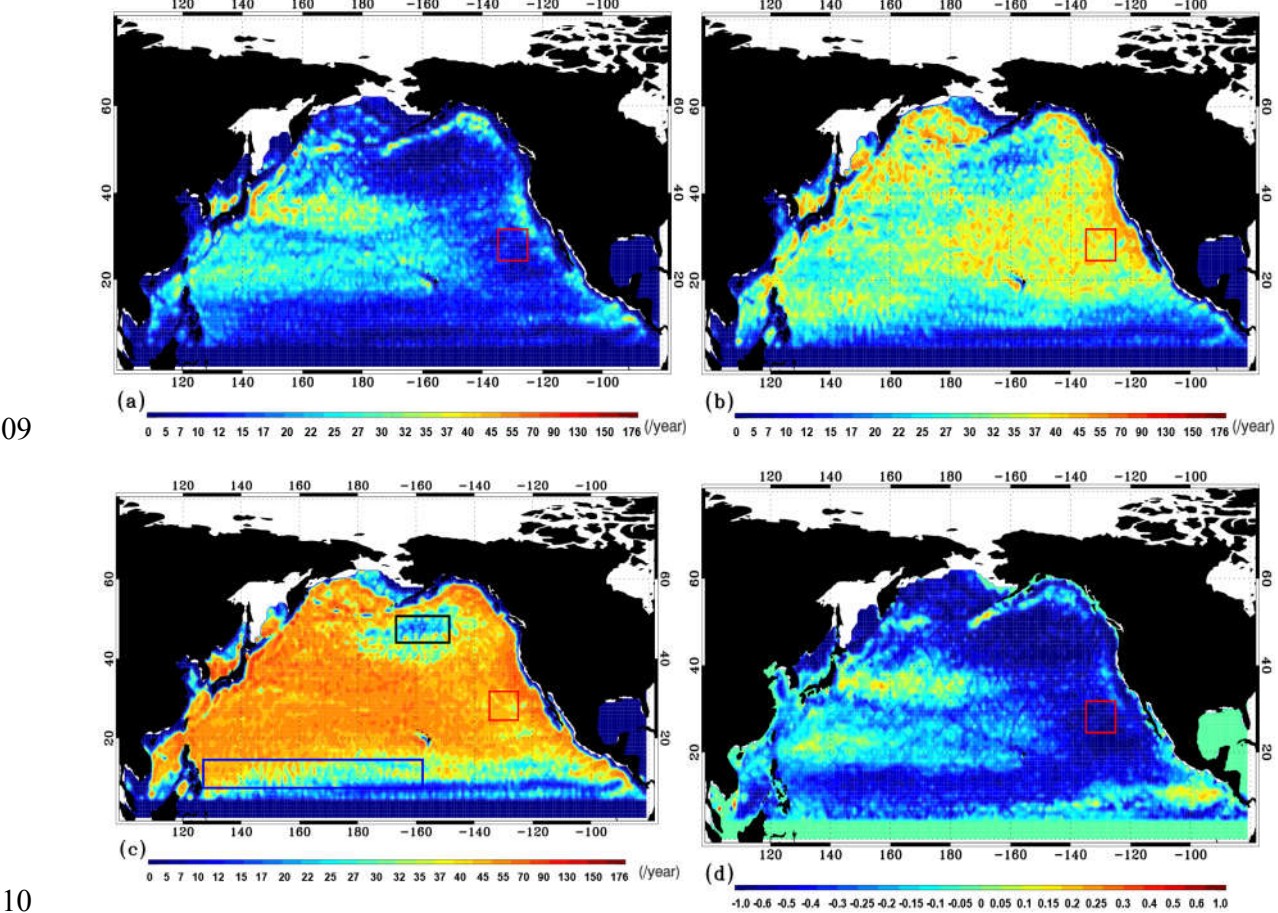

710

Figure 9 (a) The number of cyclonic eddy extrema on each $1^{\circ}\times1^{\circ}$ grid per year. (b) Same as (a), except for anticyclonic eddies. (c) Same as (a), except for the total number of eddies. (d) The ratios of difference in number of cyclonic and anticyclonic eddies to the total eddies (A logarithmic scale is used). The black box is the "eddy desert", the blue box is the NEC. The red boxes are the locations of merging/splitting examples in Figure 11, where anticyclonic eddies dominated.

716

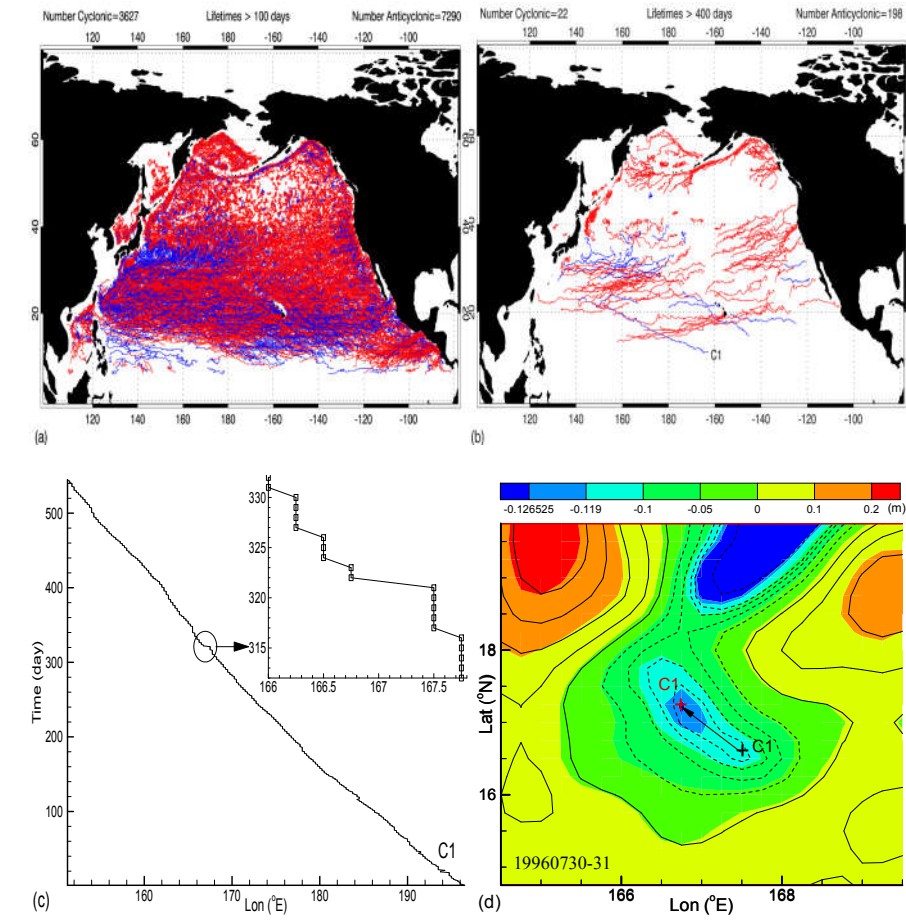

717

718

Figure 10. (a) Tracks of long-lived (>100 days) eddies. (b) Tracks of long-lived (>400 days) eddies. In (a) and (b),
blue color marks cyclonic eddies, and red color marks anticyclonic eddies. (c) The track of eddy C1. Note the
sudden jump from 167.5°E to 166.75°E on July 31, 1996. (d) The SLA fields on July 30 (contours) to 31 (shading),
using the same intervals for the contours and the shadings. The eddy centers are marked by a black cross (July 30)
and a red cross (July 31).


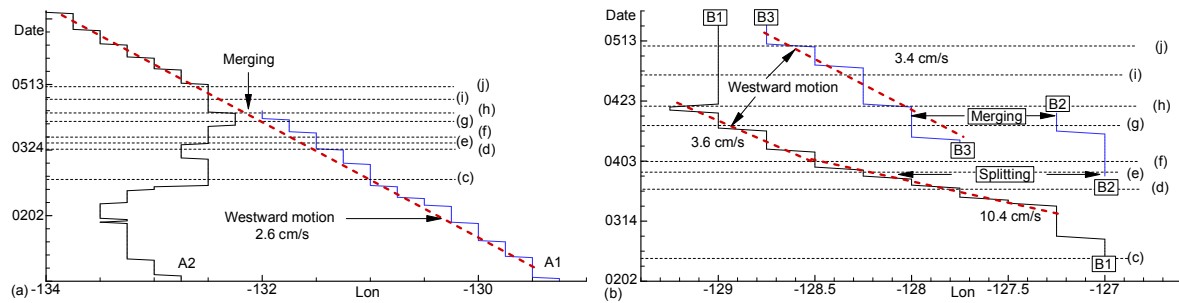



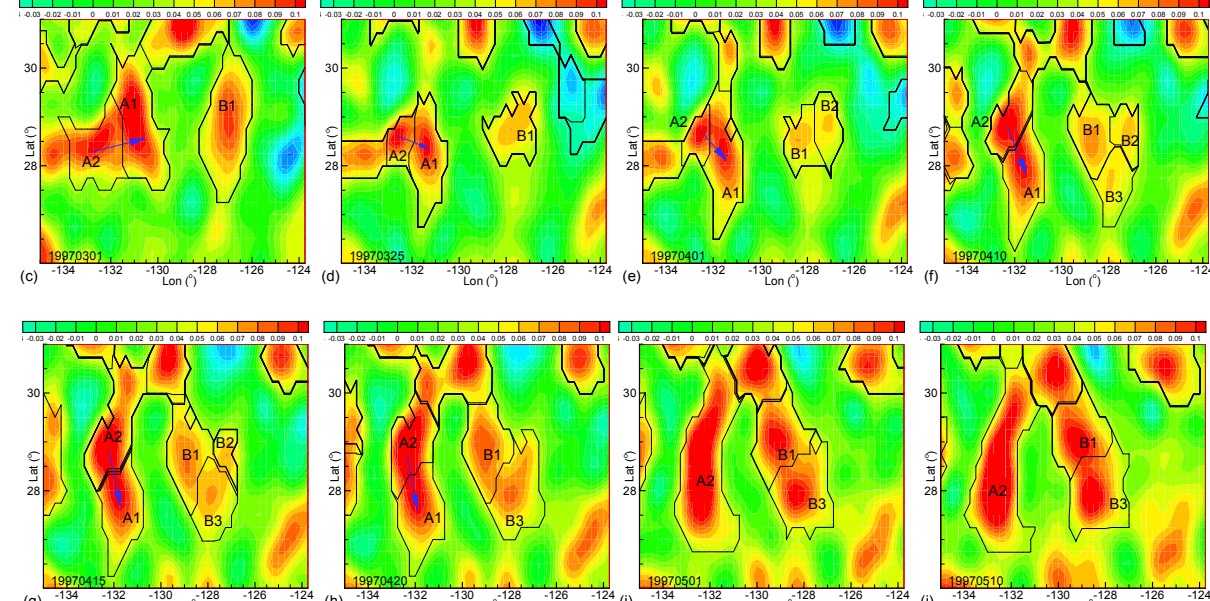


Figure 11. The dynamic evolutions of two groups of eddies, which are located in the red boxes in Fig. 9. (a) Two
eddies, A1 and A2, approached each other, and A1 merged with eddy A2, where the blue arrows indicate that the
eddy centers rotated clockwise during the merging process. (b) In the mean time, eddy B1 split into two small eddies.
(c)-(j) The evolutions of SLA fields and eddies. Note that eddies A1 and A2 had clockwise rotations when they
approached each other, as indicated by the blue arrows in (c)-(h).


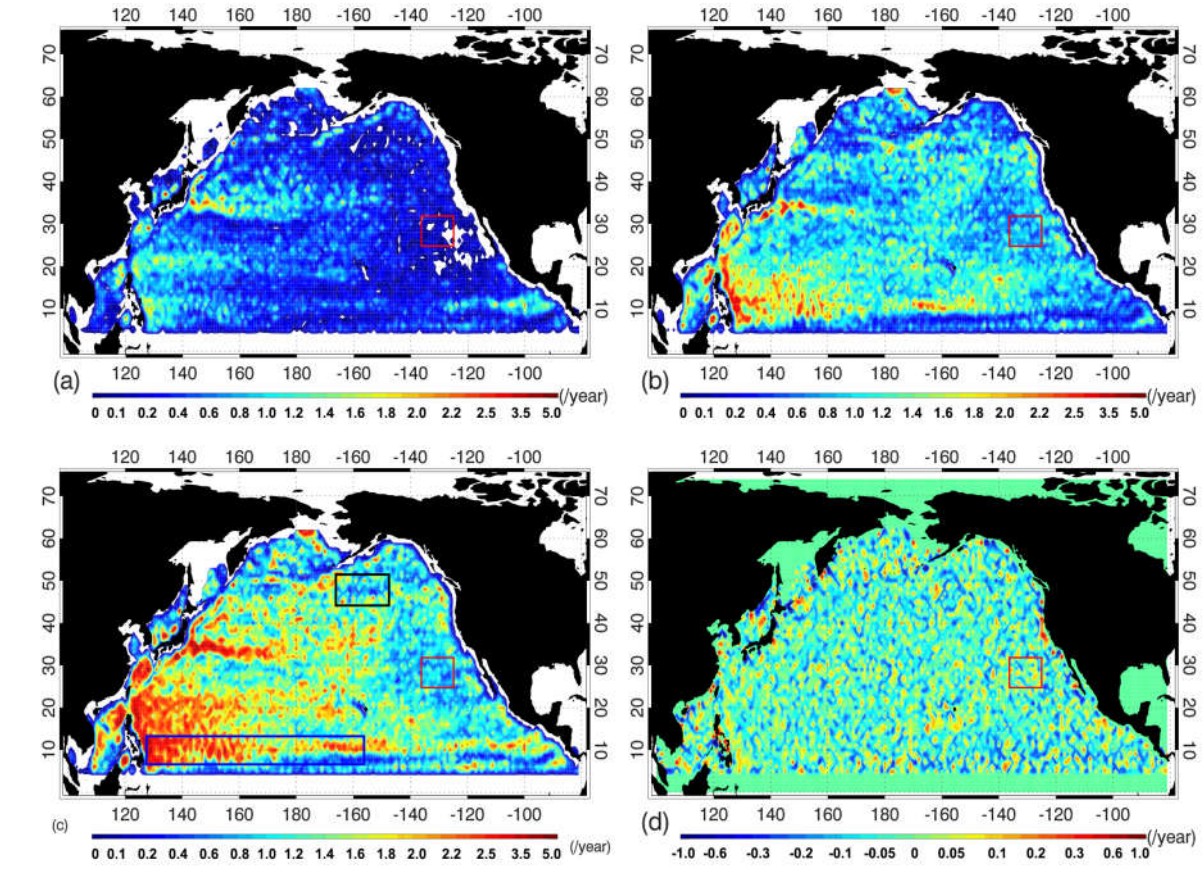



Figure 12. The frequencies of dynamic processes per 1°×1° grid element. (a) The merging frequency for cyclonic eddies. (b) The merging frequency for anticyclonic eddies. (c) The merging frequency for all eddies. (d) The ratios of difference between the frequencies of merging and splitting for all eddies to the sum of merging and splitting frequencies for all eddies. The boxes are the same in Figure 9. The blue box is the location of NEC, where merging frequency is high.

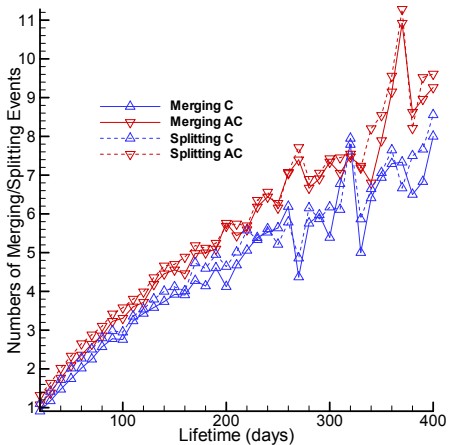


Figure 13.  The number of merging/splitting evens per eddy as function of eddy lifetime, where AC and C presents
anticyclonic and cyclonic eddies. The dynamic events are approximately linear increase with lifetime. The
merging/splitting evens are more frequent for anticyclonic eddies than for cyclonic eddies.


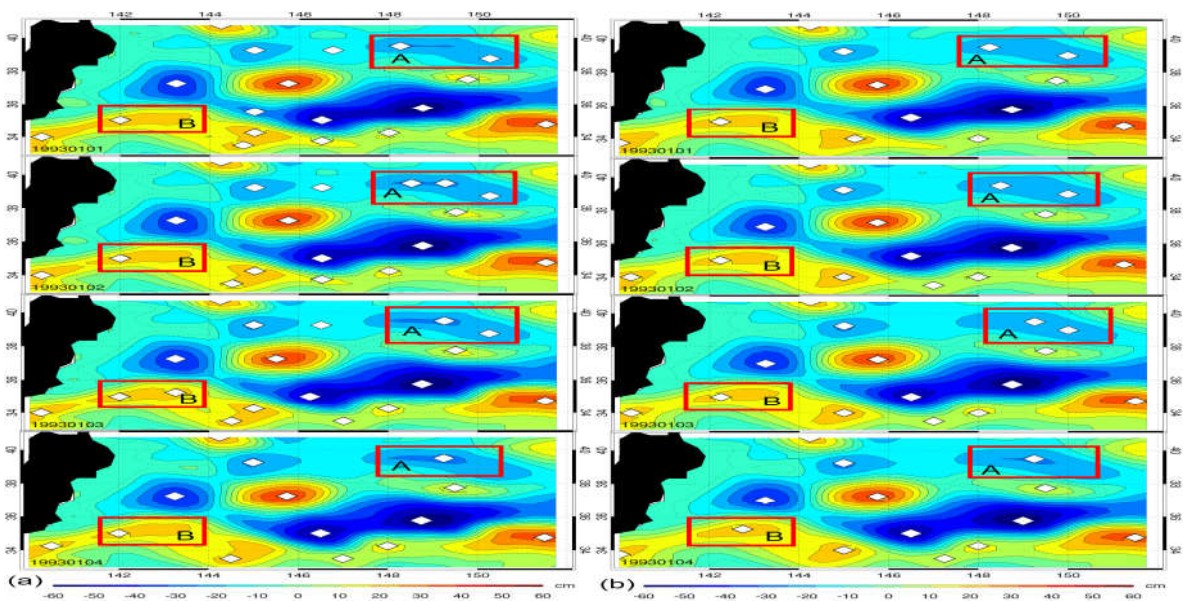


Figure 14. Comparison of the non-smoothed (a) and smoothed SLA data (b) from January 1 to January 4, 1993,
where the color field shows SLA, white dots mark eddy centers, and two boxes A and B mark the regions sensitive
to noise. Note that small noise affected the eddy detection.

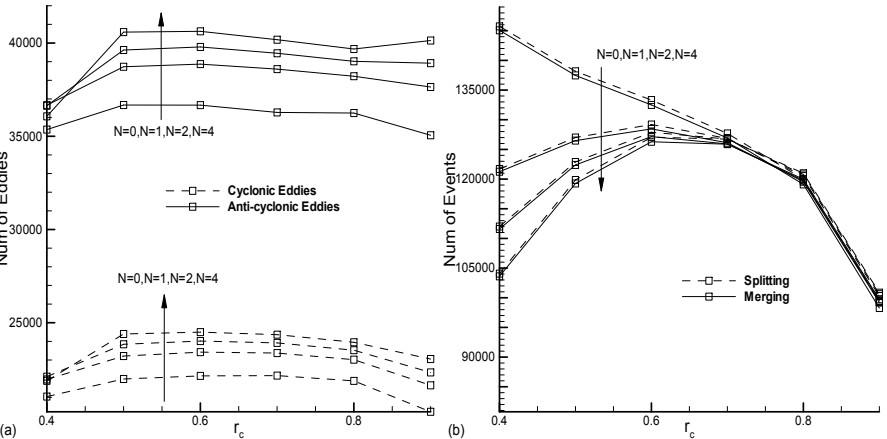

Figure 15. (a) Number of eddies (lifetime > 30 days) vs. the critical value $r_c$ and look-ahead time $N$. (b) Number of
merging and splitting events vs. the critical value $r_c$ and look-ahead time $N$.