# Peer review of "GEM: A Dynamic Tracking Model for Mesoscale Eddies in 1 the Ocean 2"

_Ocean Science, 2016_

## Referee Comment (RC1) · Anonymous Referee #1 · 31 Aug 2016

General Comments

The authors present a novel method to assess merging and splitting of ocean mesoscale eddies. The input for the associated program code are sea level anomalies. First, eddies are identified a specific way (labelled "segmentation") to allow for the assessment of merging and splitting. Second, eddies are tracked over time based on the extent of areal overlap of eddies in consecutive time steps, labelled "similarity vector". A look-ahead procedure is part of the code which improves the tracking results.

The newly developed method is unique in that it is to my knowledge the first one to address eddy merging/splitting events of eddies in a semi-Lagrangian manner and very valuable in that merging/splitting of eddies (i) poses major challenges for existing conventional eddy tracking algorithms (ii) is scientifically very interesting to investigate,

and last but not least (iii) the method is promised by the authors to be efficient, hence allowing for the processing of large datasets. I strongly recommend the publication of the paper. Given its potential great value I suggest a major effort to improve its perceptibility (see major comments below). I have been quite detailed with comments on typos/sentence structure etc- this may look like nit-picking, though my motivation was to help improve the readability of the paper to make its value easier to appreciate.

*Major comments:*
- The English is difficult to follow in some places (e.g. pay attention to sentence structure tenses), but also the technical description itself is not easy to follow in some places (noted in more detail below). Please keep in mind also that you submitted to an ocean focused journal/readership (e.g. maybe try to make the headings less technical, to use less abbreviations, to sufficiently describe the figures in the captions and to generally try to further streamline the paper).
- The first step of the algorithm, the eddy identification has been published previously as the authors say (Li et al., 2014, Li and Sun, 2015). Please state very clearly what code/section/figure you have taken straight from there, where you summarize the previous papers and where you potentially add a new facet for the purpose of the merging/splitting assessment in the present paper.
- Given that the paper presents a method and the associated code, my suggestion would be to (i) state in the text overly clearly what the input of the code is, what programming language it is written in, how efficient it is (based on the North Pacific example) and (ii) publish it along with the paper (or provide a link to a repository where you uploaded the code). Obviously, it is left to authors if they like to follow this suggestion but I believe it could greatly enhance the impact of the paper.
- I was wondering if it was useful at all to apply your tracking algorithm to eddies identified not with the watershed? Would you recommend to use the two algorithms you present to identify and track eddies "in conjunction" or does it make sense to test your tracking algorithm separately e.g. with eddies identified with OW. In the text it appears not conclusive to me, on the one hand you say that segmentation is necessary in the
identification process, on the other hand you mention in future research that one could test the algorithm with eddies identified based on other methods.

*Specific Comments:*

Text:
L14/15: isn't it mainly the look-ahead approach which solves the missing eddy problem, and the similarity approach helps to reduce the number of missing eddies?
L15 "parents and children": unclear at this point; if you want to mention this in the abstract I suggest to explain the terms.
L54 "related to sampling errors and measurement noise": I suggest to make this more general (as you do in L79) and add sth like "but also due to limitations of the eddy detection step"; a missing eddy problem can occur also in model data (e.g. if an eddy is not well defined in a time step or too weak/small etc).
L66 "larger computational complexity": I suggest to define computational complexity.
L93 "If the algorithm works well": don't you show that it works well?
L96 "to limit the size of the study area": why do you do this? Because of computational costs?
L99 "computation complex": unclear, please rephrase.
L105 "mainly include": unclear; "consists of"?
L126-129: I suggest to order the constraints the same way as you apply them in section 2.4.
L127 "the SLA value": the average SLA value?
L128: do you need the 1 cm constraint? It would be nicer without as you may discard long-lived weak eddies.
L129 "large enough": large enough for what, any reason for the 16 pixels? The "potential usefulness" noted in L136 is not obvious to me.
L138 "The above criteria...": isn't it mainly criterion (1) which helps to get rid of the huge features?

L141 "territory": any reason not to call this "area" as you have done before? In general, I suggest to stay either with "territory" or "area" unless you have a reason to use the two (e.g. also L147 "area"). The nomenclature is not always clear to me. Similarly, I would stay either with "eddy detection" or "eddy identification" if you talk about "finding" individual eddies in an SLA map (e.g. you call the section 2 "Eddy detection" and sub-section 2.2 "Eddy identification", and if you do so say early on in the paper that you will refer with "Eddy identification" always to XXX). To keep the nomenclature consistent and as simple as possible will help the reader to keep track of what you are doing. Maybe you can come up with an expression for the "final" eddies also, which includes identified eddies and their tracks (e.g. "tracked eddies")? Along these lines, what is the difference between "link" and "connection" (e.g. Fig. 4)? And, I suggest to either use "intersection" or "overlap" for the territories.

L143 ff "Necessity of segmentation": if this is taken straight from the previous Li papers I am not sure if you really need 2 Figures explaining it- I like Fig 2 to get the point across, but is Fig 3 really necessary? Also, it is somewhat unclear what is taken straight from Li (previous papers) and what is new here in terms of the eddy identification, I suggest to be very clear about it and start section 2.2 with sth like "we have taken/adopted the eddy detection step from Li et al XXX which provides us with the necessary input for the tracking routines, namely eddy territories and boundaries", followed by a summary of the method.

L143 subsection title: as you submitted to an oceanographic journal I suggest a less technical and more purpose based section title, sth like "Determination of merging and splitting events" (..."based on segmentation" if you wish).

L149 "above-identified": unclear, what do you refer to here?

L154 ff: A side note: the watershed segmentation appears elegant but the so defined territories cut across SLA contours- which appears somewhat non-physical. You mention this briefly but I would make it clearer in a sentence. If I think about trapping of mass and material properties by eddies, "stuff" in the outer area (closed SLA contours which enclose both extrema) intuitively may end up in either of the two enclosed eddies

if the split in a subsequent time step, i.e. this outer area technically does not belong to either of the two but is a separate area.

L154: it is irritating presently that 2.3 and 2.4 are named very similarly. I suggest to insert the text of 2.4 right in the beginning of 2.2 (as it provides kind of an overview), and then have subsections for the segmentation step if necessary.

L156: when do you apply constraint (4)?

L168 "which is less dependent on physical parameters": can you say why this should be a good thing? Not necessarily intuitive to me as an oceanographer.

L173 ff: this section didn't help me all that much when reading it the first time as I wasn't familiar with the technical terms used in there. I suggest to either try to make it more general/easier to understand or to get rid of it.

L177 "first": you have a "first" and "second" part here but 4 subsections which follow thereafter. I suggest to either refer first, second, third and forth here or to reduce to two subsections, if needed with subsubsections.

L178 "link between eddies in different snapshots are saved": could you define "link" here in "plain English", e.g. "link of an eddy from one temporal snapshot to the next, namely living, missing, death, birth, and the associated dynamical processes of merging and splitting?". I suggest this as after your elaboration of "map link" I still didn't get what it exactly represented.

L183 ff: I don't find it straightforward to get here the differences of "links", "branch" and "tree".

L193 "Similarity vector": once more I suggest to phrase the sections less technical and more objective driven, e.g. "Eddy similarity" or "Recognizing eddies in subsequent time steps" or so. Similarly, section 3.3 – 3.5 are titled very technically, too.

L193 ff: the section is rather long and sounds rather complicated. In my understanding, you simply define similarity based on the overlapping area of eddies in consecutive time steps. Subsequently, the overlapping area which is closest to the one of the original eddy is defined to be the successor of the original eddy (if the threshold is met). Can you summarize the section in plain English in the beginning of the section?

L195 "evaluates the similarity of these eddies": "evaluates the similarity of these eddies which is defined here based on the overlap of the territory of an eddy in two consecutive time steps." or so.

L203 "rectangular comparison region": I cannot find this in the figure.

L219 "The last type...": unclear sentence, please rephrase (e.g. "can also be identified" to "is prescribed"?).

L239 ff: I find 3.3, the look-ahead method, difficult to follow, especially paragraph 2 and 3. Partly this is due to the phrasing using T0 to T3 (rather than "unrelated" etc). Can you try to rephrase and make it clearer? Also, doesn't belong the first section more to the previous section?

L255 "closed day": I don't understand the concept of closest day, please try to elaborate.

L264: do I understand correctly that an eddy branch involves 2 time steps, and track tree is then the result of concatenating all time steps? If so, please say so, if not, please clarify. It is not entirely clear to me what makes an eddy branch.

L277 "we could not decide": why not in a technical sense? Because the similarity was not sufficiently high for either eddy?

L279 "increases": why increase, doesn't it reduce the number of detected eddies (as one has one eddy birth less as a result)?

L279 "other tracking problems": such as?

L293 "two generations": "three generations"? Parent, child and grandchild?

L294 "due to the complexity of the output": is it only that or does also the computational complexity and cost increase with more generations?

L296 "Computation complexity": "computational cost" or so? This is what I, as user of the code applying it to my data, would be interested in.

L302 This might be the fastest method possible": possible to do what? Why "might"? Change to sth like "The look-ahead method can hardly be made any faster/more efficient".

L321 "NECC": the NECC is rather close to the equator and at the boundary of your

detection domain, hence I would be careful with interpretations here.

L330 "The long-lifetime eddy trajectories imply that the quality of the tracking results is reasonable": I am not sure if this is true. You can easily connect many tracks which do not belong to each other and get a very long, but "bad" track.

L331: you look at only one example here. Can you elaborate on a few more examples and/or if you have done a visual evaluation of your tracks (e.g. animations)? One example appears not sufficient to justify the conclusion that the tracking algorithm "works".

L335 "puzzle": has this happened only once? Why did you come across it? Have you checked this somewhat systematically in your eddy data set?

L343 "A similarity vector": is it obvious if the number of missing eddies is reduced due to the fact that it is a vector or your approach of using the overlapping of territories?

L345 "as scale": "a scalar"? Btw, oftentimes, people previously have used not only distance but in addition a similarity parameter as well, but a scalar defined based on eddy properties such as amplitude/size/vorticity etc.

L364 "that cyclonic": "a pair of cyclonic"?

L365 "for cyclones": "for atmospheric cyclones"?

L369: I would be interested in the number of merging/splitting events per lifetime, could you mention it? And/or if you showed Fig 9 in terms of eddy tracks passing a grid box per year (instead of eddy extrema), one could easily compare Fig 12 to Fig 9 to roughly estimate such a number.

L388: I suggest to highlight the regions you mention in the Figure, with a box or so.

L407 "false": I don't like "false" as it is not obvious what extrema are true and what are false. The original data may have errors and hence false extrema, the smoothed data likewise.

L411 "This is one of the reasons why we need look-ahead": "The ambiguity of the eddy detection procedure and the potential errors in the input data strongly suggest the application of a look-ahead approach. The resulting eddy tracks are largely insensitive for instance to a filtering method applied to the input SLA data." Or so.

L413: in this section, it would be nice to quantify a bit more the uncertainty due to vari-

ations of N and variations of rc. For N you have partly done it (L424 where you provide the

L418 "The numbers of eddies seldom change": "The number of eddies does not change substantially". Can you quantify this a bit more? E.g. provide the

L421 "The numbers of merging and splitting events seem to converge for rc >0.5 as N increases.": "The sensitivity of the number of merging and splitting events seem to converge to 0 for rc >0.5". Can you briefly comment on why this is the case?

L425 "missing eddies, which may also reduce both total eddy numbers and dynamic events.": I would have expected the opposite (unless you consider only eddies with a certain minimum lifespan, e.g. 30 days. I guess this is the case here?)

L427 "0.6-0.7": why is this optimal?

L445 "Thus, it is difficult to directly compare the influences of eddy territory using different tracking algorithms": I don't see how this follows from what you said in the previous sentences. Also, you originally point out that you need specifically identified eddies which allow you to find the merging and splitting events. Which is why you use the watershed algorithm. Hence, overall I don't really see the point of this paragraph.

L447: The paragraph is difficult to follow except for the last two sentences. I like the sensitivity test with respect to the minimum amplitude, only an issue is that you don't strictly test the sensitivity of the eddy boundary as the modification of the amplitude threshold also changes the number of identified eddies (as you state yourself). The whole point of section 5.3 appears to be that the sensitivity of the eddy boundary is not straightforward to determine but is estimated to be small.

L457: does it make sense at all to test the tracking algorithms with the input from other detection algorithms which do not include what you refer to with "segmentation"?

L465 "A better way to obtain these parameters might be to use a nonlinear fitting of the flow field": unclear, please rephrase. The point of the whole paragraph is unclear to me.

L472 "computation": is this referring to the computation time of the tracking procedure only or including the identification? This is great. Could you repeat the detail here on

the model domain and other constraints (eddy size, amplitude, N etc) that the reader can get an impression what it would mean for his/her own application?

L475 ff: I would stress this, this is the really exiting part for me as an oceanographer.

*Figures:*

*General:*

-If you show snapshots of eddies, e.g. Fig 11 or 13, I suggest to roughly maintain an aspect ratio which doesn't distort the eddies that much (the extreme case is Fig 13!).

-please describe everything you show in your Figures, e.g. "white dots mark eddy centers" etc.; I mention some of the missing references below.

-the resolution could be increased in some of the figure and the labels increased (e.g. longitudes/latitudes are sometimes difficult to read).

Fig. 1: -mention that the background field shows SLA, and white dots mark eddy centers.

Fig. 2: -say what the letters h, A, t etc in the figure refer to.

Fig. 3: -cryptic caption, please expand (explain letters, what quantity do you illustrate in panel a etc).

Fig. 5: -missing colorbar, say what A1 etc refer to.

Fig. 6a: I suggest to shade only the overlap/intersection of eddy territories.

Fig. 6b:
-change "their are four types" to "there are four similarity types".
-in the text it says you used rc=2/3; you may want to illustrate this threshold here.
-it would help if you noted the "types of similarity" you write about in the text in the Figure in plain English, i.e. "unrelated" etc. You could also insert E2 to E4 in the respective types/quadrants to continue the illustration of 6a.

Fig. 7:
-why are the eddies labelled now Ec1 etc instead of E1 etc? If possible, make it consistent.
-cryptic caption, please expand.

Fig. 8b:
-"of the eddies" or of an example eddy?

Fig. 9:
-are all extrema counted here or only the ones belonging to eddies which existed at least 30 days?
-is the scale logarithmic or just nonlinear?
-mention the red box.
-instead of number of eddy extrema you could show number of eddy tracks per year (i.e. count an eddy which passes through only once – or have you done this? This is unclear) to make the numbers interpretable together with Fig 12.
-9c: I am a bit surprised that the eddy numbers are distributed that evenly.

Fig. 12:
-adjust colorbar so that the figure does not appear only in blueish colors.
-I cannot see the spatial pattern you describe in the text very clearly.
-12d: you could show the ratio of merging and splitting to highlight the difference (as you have done in Fig 9 for cyclones/anticyclones).

Fig. 13:
-note in the caption what are the box, letters, white dots.

Fig. 14a: I like figure 14 a lot, it is great that you tested the sensitivity of the results to the parameters.
-why do you have the spike at rc=0.6?

*Technical Corrections:*
L31 "passive": delete as you refer not only to passive tracers if I understand correctly.

L32 "has": "may have".

L39 "Recently": delete as automated ocean eddy tracking/identification has been carried out for over a decade (e.g. Isern-Fontanet et al 2003, http://dx.doi.org/10.1175/1520-0426(2003)20<772:IOMEFA>2.0.CO;2)

L42 "in any given global SLA map": insert "in any given global SLA map and they frequently interact" to connect to the following sentence.

L47 "Various": delete.

L48 "closest eddy strategy": "nearest neighbour strategy"?

L50 "searching for": "assessing" or "estimating" or so.

L58 "eddy to another": "eddy track to another".

L58 "as a result, it was abandoned, even though this look ahead feature is": "As a result, the look-ahead approach was abandoned, even though it is".

L80 "in details": "in detail".

L81 "may be more than": "may arise from more than" or so.

L82 "are merged": "subsequently merged"

L82 "it may": "$E_i$ may". Also: be consistent with using subscript or not ($E_i$, $E1$ etc).

L82 "at time step k+1": "at time step k+1 if it splits".

L90 "as daughters": "as children"?

L93 "L the pixels": "L denotes the pixels"

L94 "research field": "research fields".

L99 "examples of merging and splitting events in a sample area in the North Pacific".

L100 and others "noises": "noise".

L100 "parameters": "parameter uncertainties".

L108/109 the sentence "The data were ...": I suggest to delete this sentence. The SLA estimates are not perfect/errorless.

L111 "of gridded products": "of the gridded Aviso products".

L114 "in related studies": unclear; "depending on the purpose of a study"?

L115 "smoothing is required to remove data errors": "smoothing may be required".

L131 "especially for the small ones": delete as this is obvious with the 16pixel size

constraint.

L137 "its parameters": which parameters do you refer to?

L138 "remove": "make obsolete"?

L139 "So, they are simpler and more consistent.": unclear; delete?

L141 "to the": "to as the".

L144 "using": "based on".

L151 "seldom": "only marginally" or so.

L152 "shortened": "reduced"?

L153: I much like this approach and description!

L156 "Second...": I suggest you don't need the "second", any closed SLA contour has at least one extremum unless I am missing sth.

L156 "Third...": I suggest to say sth like "Next we apply the constraints (1) to (4)".

L158 "However, we need...": "We used a segmentation approach when applying constraint (1) [only one extremum]". Or else say why you "need" it, sth like "for the subsequent tracking procedure it is necessary to obtain eddies detected based on a segmentation approach" or so.

L161 "watershed": define/explain watershed somewhere.

L162 "slide": "tend to slide" or "would slide" or so.

L166 "to produce": I am not sure what you want to say here: I presume that you want your code to be able to do this so that the code may be more generally/widely applicable, say e.g. "and include in our code the calculation of amplitudes/etc which may be of use for oceanographic and other applications/analysis."

L167 "other": "future"?

L168 "The GEM...": I find it confusing that GEM shows up here all of a sudden; I suggest to connect to the previous paragraph with sth like "the output eddy "territory" and "boundary" from MEI is then used as input for our novel tracking algorithm GEM."

L174 "connecting": sth is missing in this sentence; connecting what?

L177 "input": "as input".

L177 "eddy data": "eddy data obtained in the prior eddy identification step".

L177 "with": delete.
L178 "different snapshots": "subsequent snapshots"?
L178 "in this part": "in this part of the work flow".
L188 "roles": do you refer with roles to parent and child or death, birth etc?
L191 "it includes": in addition to eddy territory and boundary, it needs two parameters as input".
L192 "impact of these parameter choices": "impact of varying these parameters".
L195 "As shown in Fig 5a...": to help the reader follow sth like "We first illustrate this with an example (Fig 5a) before proceeding to the mathematical expressions."
L195 "there were eddies": "there were three eddies".
L198 "territories on the first day, and on the second day": unclear; please rephrase.
L198 "the intersection was": "the intersections were"?
L198 "both": "to their respective"?
L202 "logical": "mathematically logical"?
L206 "days 0": "day 0.
L206 "we overlap the eddy territories into a single map": "we intersect the territories of day 0 and day 1"?
L211 "possibility E2": "possibility that E2".
L213 "the overlap of the same eddy territory should be large enough": unclear, please rephrase; sth like "we expect an overlap to always exist."?
L213 "one grid": "one grid box".
L213 "should be large enough.": insert a transition to the next sentence, sth like "this is one of the parameters to set" or so.
L214 "for this study": "for this study as threshold to assign E2 to E1".
L223 "way for dimensionless": "to scalar" (as oceanographer I think "unit" if I read "dimension").
L225 "good": "high".
L226 "a connection": "an evolution"?
L226 "... eddy branch procedure...": I suggest to delete this sentence, it is confusing

that you refer here to the following procedure.

L228 "However": delete.

L232 "direction": unclear, can you define/rephrase?

L236 "measure neighboring eddies": "to assess the relationship of eddies in subsequent time steps", as you refer to neighboring in time not space, don't you?

L36 "overlap in": "overlap of".

L241 "some possibilities for a give eddy": "an example for a give eddy".

L243 and others "Ec1": why do you change from simple E0, E1 etc to Ec1? Unless you want to express sth with it (e.g. it is a child of E1) I would stick with E0, E1 etc.

L243 "the eddy on day 1": "E3"?

L249 "with N daily": "with N=3"?

L250 "it is similar but slightly different than": "it is similar to the MHA method with important modifications [or differences]".

L256 "potential one": potential for what? Successor of E1?

L265 "look-ahead day and the following eddy": unclear; can you rephrase?

L266 "braches": "branches".

L280 "simply": delete.

L286 "should be": "is"?

L287 "basically": delete? Or alternatively, explain why basically?

L302: redefine M here as it is a while ago that you have done so and you define again also all the other variables.

L307 "effective": "effective compared to the previously suggested methods".

L314 "more": "more frequent".

L321 "the ratio of difference in number of": "the ratio of the difference of the numbers of".

L331 "long-life": "long-lived".

L341 "in eddy tracking used in recent studies": "applied in standard eddy tracking routines".

L342 "we should connect": "that it was consistent with our approach to connect" or "that

it was dynamically meaningful to connect" or so.

L343 "in tracking": "in the tracking procedure".

L344 "usage": unclear; "cases where the look-ahead procedure is activated" or so.

L347 "also": delete.

L355 "also occurred": "also occurred in the box the same time".

L363 "evolution process": "evolutionary process".

L371 "ubiquitous": do you mean that the events are homogeneously distributed in space? Ubiquitous and "very few times each year" sounds slightly contradictory to me.

L374 "merging frequency": "merging frequencies".

L377 "Although merging and splitting events may be ubiquitous in the ocean (Fig 12c,d), there are several types of special regions where merging and splitting events occur more frequently": the first and second part appear contradictory, how about "Although merging and splitting events may occur anywhere in the ocean there is spatial variation in the number of events." or so.

L384 "seldom": replace, with "not"? Or, if you know a study, cite the study and say "only noted by".

L388 "few": "relatively few".

L388 "The existence of "eddy desert" may be due to the fact that the eddy was too small to be detected": "One hypothesis for the existence of the eddy desert has been that eddies there were too small or weak to be detected".

L390 "However, Figures 9 and 12 suggest that fewer eddies accompanied by frequent dynamic (merging and splitting) events caused the "eddy desert."": "Figures 9 and 12 suggest that merging and splitting events may be a major contributor to the "eddy desert.""

L395 "DT14": "the Aviso product DT14".

L399 "As some parameters are used": "As certain parameters need to be chosen".

L401 "Alternatively, we can simply use a five-point quadratic smoothing to remove the noises in SLA data": "As sensitivity test we apply a simple five-point quadratic smoothing to the SLA data."

L402 "$C^2$-smooth": "$C^2$-smoothed", also please explain.

L405 "extrema": "extrema (not shown)".

L406 "These imply that the noises in DT14 data are very small.": "This implies that the noise in the DT14 data is sufficiently small for our purpose."

L413 "Impact of parameters": "Impact of variations of GEM parameters" or so.

L414 "these": which do you refer to?

L414 "apply the GEM to the eddies detected in the NPO.": "carry out a sensitivity study in the north Pacific".

L415 "the numbers": "the number".

L420 "are a little more": "occurs slightly more frequently".

L426 "the tracking results should be insensitive": "one would like the tracking results to be insensitive".

L427 "is a potential choice": "appears to be a choice with relatively robust results".

L431 "representative period": please clarify, do you refer to the temporal resolution? In this case I would say N should be larger than 4.

L432 "N=6": N=6 is not shown on the Figure.

L432 "bias": bias compared to what? E.g. "bias compared to N=XXX".

L433 "for these regimes": unclear, please rephrase.

L435 "which might to though complex evolution process": unclear, please correct sentence.

L447 "of eddy boundary": "of the eddy boundary".

L458 "GEM requires": "GEM only requires".

L460 "when considering more complex conditions.": unclear, please rephrase.

L462 "eddy parameters": I suggest to generally replace with "eddy properties" or "eddy characteristics" as you earlier refer with "parameters" to input of you model code.

L471 "increases linearly": "increases only linearly".

L473 "accompanied by other (e.g., velocity-based, O-W-based) identification methods,": delete.

L480 "for tracking": "for the tracking of the".
L484 "Each eddy track was very smooth": "Eddy tracks were smooth".
L484 "NPO": I suggest to not use this abbreviation but just say north Pacific.
L487 "observation data": "observational data".
L487 "for numerical simulation outputs": "for the output of numerical simulations".

―――――――――――――――――

---

## Referee Comment (RC2) · Anonymous Referee #2 · 6 Sep 2016

General Comments

The work presents a new eddy tracking algorithm specifically designed to account for the dynamical processes of merging and splitting combined with a look-ahead procedure.

The topic of the Ms. is of interest and relevant since although there are already other existing algorithms for eddy detection that have been proven to be robust enough, the eddy tracking problem has not been resolved adequately. The presented tracking algorithm seems to be not only a solid method, but also treats the poorly explored dynamical processes within the evolution of the eddies (merging and splitting) in an automated way. This combined with algorithms for treating missing eddies ('eddy missing problem' -look ahead for N steps- ) also adds up value to the capabilities of the developed

tracking model.

-The text is well structured and sectioned, but it has large grammatical and syntax imprecisions, which have to be corrected. Also some important concepts are not well defined in a self-explanatory way (although most of them are defined in the references).

I think the subject of the manuscript is worthy of publication and interesting for the oceanic community but some Major questions have to be addressed before a second revision round and also a careful revision of the English performed.

Major Comments

**1. There are some concepts that are used but not defined in the article (detailed in specific comments below)**

**2. I don't understand the declaration made in Section 2.2, lines 161-164 about the particles P1 and P2 in Figure 3. It is not clear what the authors refer by ''particle movements'', but I suppose that it should be particle trajectories of passive tracers that follow the streamlines of the geostrophic velocities derived from the SLA. In such case, the streamlines just follow the SLA contours, so that P1 and P2 follow the same trajectory (almost the same, because they are not exactly in the same SLA level) rotating around the 2 cyclones.**

Geostrophic velocity definition (g gravity, f coriolis parameter, H SLA) $Vx = -g/f \, dH/dy$ $Vy = g/f \, dH/dx$

Instead, it seems that you are considering that SLA is a potential of the velocity, $V = grad(H)$, like they were falling by gravity...

**3. There is missing the definition of the watershed (also pointed below).**

**4. As terminology, I would consider births and deaths within the dynamic processes, next to merging and splitting (and as opposite to kinematics). It's just a suggestion, and it's on you to do the changes.**

**5. I'm not sure that criteria 1-4 for eddy identification is what define eddies. It is clear that this defines well the maximum but regarding the ''territory" it is defined by the points inside the outmost SLA contour (as you state in line 141). So, you can have points around a maximum satisfying the conditions but being outside the defined boundary.**

**6. I think the algorithm on Similarity Vector with classification T0-T3 is very smart, but I am wondering that the variables $r1$ and $r2$ defined as fraction of overlap areas would just be valid for small time steps. I know you are using AVISO product with daily time step, but did the method work for weekly data? (note that $r1$ ($r2$) in this case would be too close to 0 (small overlaps). In this sense, I think the territory overlap may be too sensitive to the time-step, and maybe another variable (shape criteria from Mason et al. (2014) + Area, for example) could work too, being less sensitive to time-step.**

**7. Related with the previous question: in the Look-ahead, isn't the threshold $rc$ reduced? I suppose that for $N = 1$ or 2 can work with no changes, but for greater N's the ratios $r1$ $r2$ will diminish because eddy movements.**

Specific Comments

**8. Concepts that would be desirable to be defined or clarified (somewhere) in the article • Mononuclear / multinuclear eddy. I know what you are referring about, but it would be desirable to define them. • Eddy segmentation. • Watershed. How is it defined?**

**10 Page 3. L93 If the algorithms work well. . . What do you mean with this?**

**9. Page 4. L127 The SLA value of the eddy is above (below) a given SLA threshold How is defined the SLA value of the eddy? The SLA of the extremum? Also, you should give the threshold value (for sake of coherence... you are giving the threshold values that appear in conditions 3 and 4 in the next two lines).**

**10. Some (not all) grammatical issues: - look ahead/ look-ahead (sometimes**

[Figure]

with dash/sometimes without it) - straight-line / straight line model (sometimes with dash/sometimes without it) - some names are given with small letters -e.g. multiple hypothesis assignment (MHA)- and other in capital letters –e.g. Genealogical Evolution Model- . Check it for coherence

Page 1. L 32 Such transports has -> have

Page 1. L 37 Eddies can also be identified from velocity fields.

Page 3. L97 The data and eddy detection method -> methods (. . . are. . .)

Page 3. L98-99 computation complex -> complexity?

Page3. L99 illuminated -> shown

Page 4. L100 data noises -> data noise

Page 4. L105 SLA is an altimetry field, not a flow field

Page 4 L106 daily -> daily (not in cursive)

Page 4. L107-108 by the AVISO -> from AVISO

Page 4. L116-117 might affect to the eddy detection

Page 4. L120 from SLA data

Page 5.  L155 with a given threshold. . . threshold of what?  You previously defined several thresholds

Page 5. L160-164 I don't understand why do you talk about particles here, and in this way. . .

Page 6 L175 (also L190 and others) previously-identified -> previously identified

Page 7 L203 days -> day

And the list goes on.  There are many other grammatical issues that should be corrected. I am not a native English speaker but the present version of the Ms. needs to be improved.

---

## Referee Comment (RC3) · Anonymous Referee #3 · 7 Sep 2016

General Comments: I'm interesting in this paper. Compared with existing eddy tracking algorithms, the genealogical evolution model is advanced. This model is helpful to track dynamic evolution of mesoscale eddies in the ocean (especially eddy merging/splitting events), and efficiently present the eddy genealogical tree. So, I recommend this paper can be published. But I still have some questions below.

Major comments: 1. The mainly parameters are overlap rates r1 and r2, so the results will be sensitive to the temporal resolution of data (time step) and the movement speed of eddies. 1) If the time step is lager (e.g. you use AVISO weekly data), the critical value rc should be set smaller. 2) There are some different dynamic environments in NPO (e.g. background velocity in KE , STCC . . ...), the movement speed of eddies are also different. Is the constant rc properly in your model? I think the authors should discuss how to properly set the critical value rc , and pay attention to the limit conditions.

[Figure]

2. The Look-ahead approach is better and advanced. But N and rc should not be completely independent. In 5.2, "Although N=4 and N=6 might be better", is the constant rc reasonable on day 0 and day N+1 ?

Specific Comments: 3. L14 in the abstract, I was confused about 'a two-dimensional (2-D) vector' in the beginning. I thought the authors used the velocity field. Phrase similar to 'a pair of overlap rates' is simple.

4. L22∼23 in abstractïijŇthe present tense and past tense are mixed in the same sentence. Appropriate modification? E.g. < Although eddy splitting and merging are ubiquitous in the ocean, they have different geographic distribution in the Northern Pacific Ocean. Both the merging and splitting rates of the eddies are high, especially at the western boundary, along major currents and in "eddy deserts."> I am also not a native English speaker. please refer to other reviewers about the grammatical issues.

5. L489 In the Conclusion, <"MEI" and "GEM" computer codes. . ...>. Can the authors add some sentences about the codes? E.g.The code language is matlab, fortran or C? how to reserve/save the genealogy tree information in figure8b ?

———————————————

---

## Author Comment (AC1) · 4 Oct 2016

Major comments: 1. The mainly parameters are overlap rates r1 and r2, so the results will be sensitive to the temporal resolution of data (time step) and the movement speed of eddies. 1) If the time step is lager (e.g. you use AVISO weekly data), the critical value rc should be set smaller. 2) There are some different dynamic environments in NPO (e.g. background velocity in KE , STCC $: : :..$), the movement speed of eddies are also different. Is the constant rc properly in your model? I think the authors should discuss how to properly set the critical value rc , and pay attention to the limit conditions.

Reply: Yes, we agree that the critical value rc should be depended on time step, and we simply use a constant rc here just for illustrate the model. Users can use a non-constant rc as their will. We add the point in discuss section 5.2.

2. The Look-ahead approach is better and advanced. But N and rc should not be completely independent. In 5.2, "Although N=4 and N=6 might be better", is the constant rc reasonable on day 0 and day N+1 ?

Reply: We simply use a constant rc. Although N and rc should be something relative, we are not clear how to optimally deal with it. For example, one can use rc=rc_0-a*N. But others may comment how to choose a, or even suspect this relationship.

Specific Comments:

3. L14 in the abstract, I was confused about 'a two-dimensional (2-D) vector' in the beginning. I thought the authors used the velocity field. Phrase similar to 'a pair of overlap rates' is simple.

Reply: Thanks for this suggestion. Phrase 'a pair of overlap rates' is simple, but may also bring other problems. For example, it is hard to descript Fig 6b using 'a pair of overlap rates'. Maybe we should modify as "a two-dimensional (2-D) similarity vector".

4. L22∼23 in abstract ïijNthe present tense and past tense are mixed in the same sentence. Appropriate modification? E.g. < Although eddy splitting and merging are ubiquitous in the ocean, they have different geographic distribution in the Northern Pacific Ocean. Both the merging and splitting rates of the eddies are high, especially at the western boundary, along major currents and in "eddy deserts."> I am also not a native English speaker. please refer to other reviewers about the grammatical issues.

Reply: Thanks for your suggestion. We will change them according to all reviewers' comments.

5. L489 In the Conclusion, <"MEI" and "GEM" computer codes$: : :..$>. Can the authors add some sentences about the codes? E.g. The code language is matlab, fortran or C? how to reserve/save the genealogy tree information in figure8b?

Reply: The codes are written in Fortran 90/95. We save part of the tree information (There are more information not output can be found in the codes, users can output them as their will.) with "Eddy-Eddr.dat", which contains the parent/child eddies (if it has) and merging/splitting events (if it has). One can read the manual for this information.

---

## Author Comment (AC2) · 4 Oct 2016

Major Comments

**1. There are some concepts that are used but not defined in the article (detailed in specific comments below)**

**2. I don't understand the declaration made in Section 2.2, lines 161-164 about the particles P1 and P2 in Figure 3. It is not clear what the authors refer by ''particle movements'', but I suppose that it should be particle trajectories of passive tracers that follow the streamlines of the geostrophic velocities derived from the SLA. In such case, the streamlines just follow the SLA contours, so that P1 and P2 follow the same trajectory (almost the same, because they are not exactly in the same SLA level) rotating around the 2 cyclones. Geostrophic velocity definition (g gravity, f coriolis parameter, H SLA) $V_x = -g/f \, dH/dy$ $V_y = g/f \, dH/dx$ Instead, it seems that you are considering that SLA is a potential of the velocity, $V = grad(H)$, like they were falling by gravity...**

Reply: Yes, they were falling by gravity, which is the origin of watershed idea.

**3. There is missing the definition of the watershed (also pointed below).**

Reply: We add the notation in section 2.4. (For basins, the "watershed" is a ridge between them, while it is a valley for plateaus).

**4. As terminology, I would consider births and deaths within the dynamic processes, next to merging and splitting (and as opposite to kinematics). It's just a suggestion, and it's on you to do the changes.**

Reply: We are not clear this comment. In our codes, we treat them equally. The user can have their own strategies and modify the codes, if he wants.

**5. I'm not sure that criteria 1-4 for eddy identification is what define eddies. It is clear that this defines well the maximum but regarding the ''territory'' it is defined by the points inside the outmost SLA contour (as you state in line 141). So, you can have points around a maximum satisfying the conditions but being outside the defined boundary.**

Reply: We are not clear what you concern. We have descript the algorithm of identification in section 2.4 (*Now, we rewrite this part based on comments by you and others*). Since we find a simply-connected region with a given threshold (see reply #9 below), it seems that no points outside the defined boundary could be taken as eddy.

**6. I think the algorithm on Similarity Vector with classification T0-T3 is very smart, but I am wondering that the variables r1 and r2 defined as fraction of overlap areas would just be valid for small time steps. I know you are using AVISO product with daily time step, but did the method work for weekly data? (note that r1 (r2) in this case would be too close to 0 (small overlaps). In this sense, I think the territory overlap may be too sensitive to the time-step, and maybe another variable (shape criteria from Mason et al. (2014) + Area, for example) could work too, being less sensitive to time-step.**

Reply: We totally agree with you that ratios r1 and r2 would just be valid for small time steps. We also agree that another variable (shape criteria from Mason et al. (2014) + Area, for example) could also work too, which we have mentioned as alternative way for dimensionless similarity parameters. But we think time-step should be considered as limitation. This is something like CFL condition (for time step) in

computer fluid dynamics. And we think any tracking method should have this time-step limitation (depending on eddy size/propagation speed), if one don't want to mix one signal with another. We add this point in discussion section 5.2.

**7. Related with the previous question: in the Look-ahead, isn't the threshold rc reduced? I suppose that for N = 1 or 2 can work with no changes, but for greater N's the ratios r1 r2 will diminish because eddy movements.**

Reply: We totally agree with you that ratios r1 and r2 will diminish for greater N's. A general treatment seems to be useful, but we do not know the accrue relationship. For example, one can use rc=rc_0-a*N. But others may comment how to choose a, or even suspect this relationship. So in this paper, we use only small N's in the approach.

Specific Comments

**8. Concepts that would be desirable to be defined or clarified (somewhere) in the article âA˘ c Mononuclear / multinuclear eddy. I know what you are referring about, but ´it would be desirable to define them. âA˘ c Eddy segmentation. â´A˘ c Watershed. How is ´it defined?**

Reply: we now add the definition of Mononuclear / multinuclear eddy in section 2.2. "The eddies may be identified with multinuclear (two or more SLA extremes in one eddy) or mononuclear (only one SLA extremum in one eddy)".

**10 Page 3. L93 If the algorithms work well: : : What do you mean with this?**

Reply: We are sorry for the unclear. We mean, if the algorithms was implemented with the computer codes properly. Since the GEM model itself (Fig 8) doesn't take efficient algorithms/codes for granted, it can be implemented with different algorithms/codes by users. Thus, the accurate and efficient may be much different. We have modified to "if the GEM was implemented with the computer codes properly".

**9. Page 4. L127 The SLA value of the eddy is above (below) a given SLA threshold How is defined the SLA value of the eddy? The SLA of the extremum? Also, you should give the threshold value (for sake of coherence... you are giving the threshold values that appear in conditions 3 and 4 in the next two lines).**

Reply: We choose 3 cm as threshold in our calculation. The potential cyclonic eddy should be within a simply-connected region whereas SLA>3 cm. The anticyclonic eddy SLA<-3 cm.

**10. Some (not all) grammatical issues: - look ahead/ look-ahead (sometimes with dash/sometimes without it) - straight-line / straight line model (sometimes with dash/sometimes without it) - some names are given with small letters -e.g. multiple hypothesis assignment (MHA)- and other in capital letters –e.g. Genealogical Evolution Model- . Check it for coherence**

Reply: Thanks.

Page 1. L 32 Such transports has -> have

Reply: Thanks.

Page 1. L 37 Eddies can also be identified from velocity fields.

Reply: Thanks.

Page 3. L97 The data and eddy detection method -> methods (∴∴ are∴∴)

Reply: Thanks.

Page 3. L98-99 computation complex -> complexity?

Reply: Thanks, we modified to complexity.

Page3. L99 illuminated -> shown

Reply: Thanks.

Page 4. L100 data noises -> data noise

Reply: Thanks.

Page 4. L105 SLA is an altimetry field, not a flow field

Reply: Thanks.

Page 4 L106 daily -> daily (not in cursive)

Reply: Thanks.

Page 4. L107-108 by the AVISO -> from AVISO

Reply: Thanks.

Page 4. L116-117 might affect to the eddy detection

Reply: Thanks.

Page 4. L120 from SLA data

Reply: Thanks.

Page 5. L155 with a given threshold∴∴ threshold of what? You previously defined several thresholds

Reply: The threshold of SLA >3 cm for cyclonic eddies, SLA<-3 cm for anticyclonic eddies.

Page 5. L160-164 I don't understand why do you talk about particles here, and in this way∴∴

Reply: Since the idea of watershed requires that particles move by gravity. This naturally provides a way to segment the eddy.

Page 6 L175 (also L190 and others) previously-identified -> previously identified

Reply: Thanks.

Page 7 L203 days -> day

Reply: Thanks.

---

## Editor Comment (EC1) · J. M. Huthnance (Editor) · 13 Oct 2016

Dear Authors

Below is a quote from the Ocean Science Web site, in answer (I hope) to your question about a .gif movie.

Yours sincerely John Huthnance

Any supplementary material (if available) must be submitted as a *.zip archive or single *.pdf file. The overall file size of a supplement is limited to 50 MB. Authors of larger supplements are kindly asked to submit their files to a reliable data repository and to insert a link in the manuscript. Ideally, this linkage is realized through DOIs (digital object identifiers).

---

## Author Comment (AC3) · 13 Oct 2016

1 GEM is relatively independent to eddy identification MEI, but the ratios r1 and r2 might be sensitive to the method used in identification. We noted that O-W based identification is much sensitive than SLA based one, since O-W based eddies are much smaller. We also point out that ratios r1 and r2 should only be valid for small time steps. This is something like CFL condition (for time step) in computer fluid dynamics. In general, we think any tracking method should have this time-step limitation (depending on eddy size/propagation speed), if one don't want to mix one signal with another.

2 There are lots of merging/splitting examples according to the data. We made some animations (.gif files) of them, but we do not want to include them into the present paper except for the example in Fig.11, because the paper is too long to be hold by us. From

2014, we spend too much time in reversing this paper but few time in the analysis of results and other studies. Besides, one technical problem for us is how to illuminate it since .gif file can't play in word. Should we take the animation as supplement?

3 Without segmentation, the strong interaction of eddies "in conjunction", which leads to genesis and termination of eddies, is more likely missed. However, some weak interaction of eddies in some far distance (watershed free) could still be recorded. We are sure for this because we have analysized many cases of the output data. We noted (not mentioned in this paper) that lots of merging/splitting records occurred at the interaction of two eddies with a certain distance. This kind of interaction can't be recorded by previously interaction-free tracking algorithm (only isolated tracking eddy record), but it is still scientifically very interesting to investigate.

4 We add a new figure 13, which presents the merging/splitting events with lifetime.

5 The codes are written in Fortran 90/95 standard with windows platform, including seven f90 program files (2 for MEI, 4 for GEM and 1 for both). In order to maintain the potential reversions of the codes and manuals, we plan to upload both the codes and manual at https://www.researchgate.net/profile/Liang_Sun20/ after the publication of the paper.

---

## Author Comment (AC4) · 14 Oct 2016

Thanks for your efforts and useful comments. We make a summary of brief reply to the major comments by all reviewers. Since your comments are so many, we make a point to point reply to your major, specific and figure comments. The technical corrections (typos, grammars) are made simply by following the suggestions without any reply.

*Major comments:*

- The first step of the algorithm, the eddy identification has been published previously as the authors say (Li et al., 2014, Li and Sun, 2015). Please state very clearly what code/section/figure you have taken straight from there, where you summarize the previous papers and where you potentially add a new facet for the purpose of the merging/splitting assessment in the present paper.

**Reply**: We rewrite section 2.2 based on comments by you and others.

- Given that the paper presents a method and the associated code, my suggestion would be to (i) state in the text overly clearly what the input of the code is, what programming language it is written in, how efficient it is (based on the North Pacific example) and (ii) publish it along with the paper (or provide a link to a repository where you uploaded the code). Obviously, it is left to authors if they like to follow this suggestion but I believe it could greatly enhance the impact of the paper.

**Reply**: The programs are written in Fortran 90/95 standard with windows platform, including seven f90 program files. In a personal computer with CPU of i7-6700k and 4.00 GHz, program Mei.f90 uses 15 minutes to identify vortices, program Vortex.f90 uses about 25 minutes to establish similarity, and program Eddy.f90 uses about 10 minutes to track eddies. We state these in our program readme file, since lots of program details may not be concerned by users or readers. We simply copy the relative part below.

Data and file preparation

GEM uses SLA data as input. We store all the SLA data files ( 'msla_19930101.dat' ) in a given directory as "SLApath", and all the data filename (19930101, 19930102, etc) in a txt file ('intxt =E:\sunl\WORK\Eddytrack\ini.txt'). Besides, we also use a topography file ("topofile=E:\sunl\WORK\Eddytrack\topo_721_1440.txt") since the SLA data have useless values on lands or coastal regions.

Then we need to prepare initial parameter file of 'inipara.txt' (see inipara in "Typeconst3.f90"), which includes these parameters.

```
open(22,file=inipara,form='formatted',action='read')
read(22,*) iconmin,iconmax, ias,ide !
read(22,*) intxt,topofile      ! datefile, topofile
read(22,*) SLApath              ! SLA data path
read(22,*) vorpath,eddypath     ! vortices path, eddy path
close(22)
```

where iconmin=1 (first file), iconmax=7305 (last file of 20 year SLA data), ias=ide=1 is used for similarity calculations in "vortex.f90".

The outputs of programs include identified vortices (vorpath) from SLA data and eddy tracks from vortices (eddypath).

In "vorpath", there are three series of files as "date-vor.dat", "date-voi.dat", "date-vet.dat", where "date" means "19930101, 19930102, etc."

In " eddypath", there are "Eddy-eddv.dat" and "Eddy-eddt.dat" (Eddy is identified as a integer number starting from 1000000), which contains long-term eddies (defined as life time>$MP\_Eddylif\_CV\_MIN$). Besides there are two files as "TbEddy.txt" and "TpEddy.txt", which list parameters of all eddies (even life time< $MP\_Eddylif\_CV\_MIN$).

- I was wondering if it was useful at all to apply your tracking algorithm to eddies identified not with the watershed? Would you recommend to use the two algorithms you present to identify and track eddies "in conjunction" or does it make sense to test your tracking algorithm separately e.g. with eddies identified with OW. In the text it appears not conclusive to me, on the one hand you say that segmentation is necessary in the identification process, on the other hand you mention in future research that one could test the algorithm with eddies identified based on other methods.

**Reply**: In our programs, user can use either OW, GV (Geostrophic Vorticity, normalized by Coriollis parameter $f$), SLA, or hybrid of them to identify eddy. But a watershed algorithm is always used in these programs. The identified eddies by using other identification algorithm without watershed can also be tracked with our programs given proper input. In this case, there should be no difference in eddy track itself (Eddy-Eddv.dat). But the records of merging /splitting relationship (Eddy-Eddr.dat) will change. The strong interaction of eddies "in conjunction", which leads to genesis and termination of eddies, is more likely missed. However, some weak interaction of eddies in some far distance (watershed free) could still be recorded. We are sure for this because we have analysized many cases of our output data. We noted (not mentioned in this paper) that lots of merging/splitting records occurred at the interaction of two eddies with a certain distance. This kind of interaction can't be recorded by previously interaction-free tracking algorithm (only isolated tracking eddy record), but it is still scientifically very interesting to investigate. Finally, the programs at least have "look-ahead" approach, which can effectively reduce the temporarily missing eddies at identification comparing with previous programs.

*Specific Comments:*
Text:
Comment L14/15: isn't it mainly the look-ahead approach which solves the missing eddy problem, and the similarity approach helps to reduce the number of missing eddies?
**Reply**: Yes, both approaches solve different problems in missing eddies, e.g., the similarity approach reduce the mistakes of one eddy track to another.
Comment L15 "parents and children": unclear at this point; if you want to mention this in the abstract I suggest to explain the terms.
**Reply**: suggestion followed. "parents (a new eddy) and children (e.g., splitting eddies from parent eddy)"
Comment L54 "related to sampling errors and measurement noise": I suggest to make this more general (as you do in L79) and add sth like "but also due to limitations of the eddy detection step"; a missing eddy problem can occur also in model data (e.g. if an eddy is not well defined in a time step or too weak/small etc).
**Reply**: suggestion followed.
Comment L66 "larger computational complexity": I suggest to define computational

complexity.

**Reply**: suggestion followed.

Comment L93 "If the algorithm works well": don't you show that it works well?

**Reply**: We are sorry for the unclear. We mean, if the algorithms was implemented with the computer codes properly. Since the GEM model itself (Fig 8) doesn't take efficient algorithms/codes for granted, it can be implemented with different algorithms/codes by users. Thus, the accurate and efficient may be much different. We have modified to "if the GEM was implemented with the computer codes properly".

Comment L96 "to limit the size of the study area": why do you do this? Because of computational costs?

**Reply**: Since we are more familiar with NPO, we only output NPO data to limit the analysize region. However the codes are for global oceans.

Comment L99 "computation complex": unclear, please rephrase.

**Reply**: Suggestion followed: "computational complexity".

Comment L105 "mainly include": unclear; "consists of"?

**Reply**: Suggestion followed " consists of"

Comment L126-129: I suggest to order the constraints the same way as you apply them in section 2.4.

**Reply**: Suggestion followed. We rearrange it in section 2.2

Comment L127 "the SLA value": the average SLA value?

**Reply**: No. All the SLA values.

Comment L128: do you need the 1 cm constraint? It would be nicer without as you may discard long-lived weak eddies.

**Reply**: Yes, we used this 1 cm constraint so that weak extremes are not taken as eddies.

Comment L129 "large enough": large enough for what, any reason for the 16 pixels? The "potential usefulness" noted in L136 is not obvious to me.

**Reply**: large enough for estimating eddy parameters.

Comment L138 "The above criteria...": isn't it mainly criterion (1) which helps to get rid of the huge features?

**Reply**: Yes, criterion (1) which helps to get rid of some features. Then criteria (3) and (4) also do this job.

Comment L141 "territory": any reason not to call this "area" as you have done before? In general, I suggest to stay either with "territory" or "area" unless you have a reason to use the two (e.g. also L147 "area"). The nomenclature is not always clear to me. Similarly, I would stay either with "eddy detection" or "eddy identification" if you talk about "finding" individual eddies in an SLA map (e.g. you call the section 2 "Eddy detection" and subsection 2.2 "Eddy identification", and if you do so say early on in the paper that you will refer with "Eddy identification" always to XXX). To keep the nomenclature consistent and as simple as possible will help the reader to keep track of what you are doing. Maybe you can come up with an expression for the "final" eddies also, which includes identified eddies and their tracks (e.g. "tracked eddies")? Along these lines, what is the difference between "link" and "connection" (e.g. Fig. 4)? And, I suggest to either use "intersection" or "overlap" for the territories.

**Reply**: We use area in this paper. The "map link" is relation between two time steps and then

connecting all time steps to the "track tree."

Comment L143 ff "Necessity of segmentation": if this is taken straight from the previous Li papers I am not sure if you really need 2 Figures explaining it- I like Fig 2 to get the point across, but is Fig 3 really necessary? Also, it is somewhat unclear what is taken straight from Li (previous papers) and what is new here in terms of the eddy identification, I suggest to be very clear about it and start section 2.2 with sth like "we have taken/adopted the eddy detection step from Li et al XXX which provides us with the necessary input for the tracking routines, namely eddy territories and boundaries", followed by a summary of the method.

**Reply**: we add the sentence to section 2.2

Comment L143 subsection title: as you submitted to an oceanographic journal I suggest a less technical and more purpose based section title, sth like "Determination of merging and splitting events" (..."based on segmentation" if you wish).

**Reply**: Suggestion followed.

Comment L149 "above-identified": unclear, what do you refer to here?

**Reply**: The amplitude and area of eddies.

Comment L154 ff: A side note: the watershed segmentation appears elegant but the so defined territories cut across SLA contours- which appears somewhat non-physical. You mention this briefly but I would make it clearer in a sentence. If I think about trapping of mass and material properties by eddies, "stuff" in the outer area (closed SLA contours which enclose both extrema) intuitively may end up in either of the two enclosed eddies if the split in a subsequent time step, i.e. this outer area technically does not belong to either of the two but is a separate area.

**Reply**: We rewrite this part with the order of identification by following the suggestion before and hope it is more clear now.

Comment L154: it is irritating presently that 2.3 and 2.4 are named very similarly. I suggest to insert the text of 2.4 right in the beginning of 2.2 (as it provides kind of an overview), and then have subsections for the segmentation step if necessary.

**Reply**: We insert first paragraph of 2.4 right after 2.2, and combine 2.3 and 2.4 as one section.

Comment L156: when do you apply constraint (4)?

**Reply**: In the last step, we used this 1 cm and 16 pixels constraint so that weak/small extremes are not taken as eddies. We add this to section 2.4. And we rewrite the paragraph.

Comment L168 "which is less dependent on physical parameters": can you say why this should be a good thing? Not necessarily intuitive to me as an oceanographer.

**Reply**: GEM itself should be less dependent on physical parameters. However, MEI does not. It is MEI that defines the eddy with physical parameters. And we found that GEM parameters are sensitive to the eddy identification method. We add a paragraph in discussion section 5.2.

Comment L173 ff: this section didn't help me all that much when reading it the first time as I wasn't familiar with the technical terms used in there. I suggest to either try to make it more general/easier to understand or to get rid of it.

**Reply**: We use subsubsections to do so.

Comment L177 "first": you have a "first" and "second" part here but 4 subsections which follow thereafter. I suggest to either refer first, second, third and forth here or to reduce to two subsections, if needed with subsubsections.

**Reply**: We use subsubsections to do so.

Comment L178 "link between eddies in different snapshots are saved": could you define "link" here in "plain English", e.g. "link of an eddy from one temporal snapshot to the next, namely living, missing, death, birth, and the associated dynamical processes of merging and splitting?". I suggest this as after your elaboration of "map link" I still didn't get what it exactly represented.

**Reply**: The relationships are determined by two relatively independent steps i.e. the GEM algorithm consists of two parts (see Fig.4 for details): first, measuring the "map link" between two time steps and then connecting all time steps to the "track tree."

Comment L183 ff: I don't find it straightforward to get here the differences of "links", "branch" and "tree".

**Reply**: an eddy branch involves 2 time steps, and track tree is then the result of concatenating all time steps

Comment L193 "Similarity vector": once more I suggest to phrase the sections less technical and more objective driven, e.g. "Eddy similarity" or "Recognizing eddies in subsequent time steps" or so. Similarly, section 3.3 – 3.5 are titled very technically, too.

**Reply**: We modify to Eddy similarity

Comment L193 ff: the section is rather long and sounds rather complicated. In my understanding, you simply define similarity based on the overlapping area of eddies in consecutive time steps. Subsequently, the overlapping area which is closest to the one of the original eddy is defined to be the successor of the original eddy (if the threshold is met).

Can you summarize the section in plain English in the beginning of the section?

**Reply**: we summarize this at the beginning of the section.

Comment L195 "evaluates the similarity of these eddies": "evaluates the similarity of these eddies which is defined here based on the overlap of the territory of an eddy in two consecutive time steps." or so.

**Reply**: suggestion followed.

Comment L203 "rectangular comparison region": I cannot find this in the figure.

**Reply**: We do not show this in figure, because the rectangular may .

Comment L219 "The last type...": unclear sentence, please rephrase (e.g. "can also be identified"to "is prescribed"?).

**Reply**: suggestion followed.

Comment L239 ff: I find 3.3, the look-ahead method, difficult to follow, especially paragraph 2 and 3. Partly this is due to the phrasing using T0 to T3 (rather than "unrelated" etc). Can you try to rephrase and make it clearer? Also, doesn't belong the first section more to the previous section?

**Reply**: We modify the figure and add notations on each type, e.g. T3(living). For example,

Comment L255 "closed day": I don't understand the concept of closest day, please try to elaborate.

**Reply**: Maybe "earliest day".

Comment L264: do I understand correctly that an eddy branch involves 2 time steps, and track tree is then the result of concatenating all time steps? If so, please say so, if not, please clarify. It is not entirely clear to me what makes an eddy branch.

**Reply**: Yes, as you mentioned: an eddy branch involves 2 time steps, and track tree is then the result of concatenating all time steps.

Comment L277 "we could not decide": why not in a technical sense? Because the similarity was not sufficiently high for either eddy?

**Reply**: Yes, modified.

Comment L279 "increases": why increase, doesn't it reduce the number of detected eddies (as one has one eddy birth less as a result)?

**Reply**: This choice (keeping parent eddies $P_1$ and $P_2$ alive) artificially increases lifetimes of eddy P1 and P2.

Comment L279 "other tracking problems": such as?

**Reply**: This can't be addressed in a few words, but readers can image what would happen. At first, in each time step, we need write the same track to two different eddies. To this end, in each time step, we need to check each eddy whether it has a company eddy, and find out which eddy number is, where it is stored, etc. This will make the codes very complex and would easily have bugs in codes. Besides, the storage can't support this, if this new eddy has merging/splitting events eventually.

Comment L293 "two generations": "three generations"? Parent, child and grandchild?

**Reply**: We only records two generations "Parent and child", due to the storage and complexity. The grandchild is recorded as the child of a child eddy.

Comment L294 "due to the complexity of the output": is it only that or does also the computational complexity and cost increase with more generations?

**Reply**: The computational complexity and cost have limited increase. But the logical complexity, store structure and the codes increase very much.

Comment L296 "Computation complexity": "computational cost" or so? This is what I, as user of the code applying it to my data, would be interested in.

**Reply**: In computer science, it is "computation complexity". So we simply follow this.

Comment L302 This might be the fastest method possible": possible to do what? Why "might"? Change to sth like "The look-ahead method can hardly be made any faster/more efficient".

**Reply**: modified as suggestion.

Comment L321 "NECC": the NECC is rather close to the equator and at the boundary of your detection domain, hence I would be careful with interpretations here.

**Reply**: Thanks for suggestion. It is NEC but NECC.

Comment L330 "The long-lifetime eddy trajectories imply that the quality of the tracking results is reasonable": I am not sure if this is true. You can easily connect many tracks which do not belong to each other and get a very long, but "bad" track.

**Reply**: If the eddy satisfy the overlap criteria, the track will be ok. Otherwise, there is a risk of connecting many tracks which do not belong to each other, especially by those simple tracking algorithms.

Comment L331: you look at only one example here. Can you elaborate on a few more examples and/or if you have done a visual evaluation of your tracks (e.g. animations)? One example appears not sufficient to justify the conclusion that the tracking algorithm "works".

**Reply**: Yes, we have many such of examples and made animations of them. We can send one or two of them to editor. But we do not want to include them into the present paper except for the example in Fig.11, because the paper is too long to be hold by us. From 2014, we spend too much time in reversing this paper but few time in the analysis of results and other studies.

Comment L335 "puzzle": has this happened only once? Why did you come across it? Have you checked this somewhat systematically in your eddy data set?

**Reply**: For this example, it is only once. We noted such kind of jump before and found that is hard to choose distance criteria in tracking. This is the reason why we want to develop a new tracking method and solve the problem from beginning. But for this example, it is only a chance. We only drew the long-term eddy track, and found that there is a jump. We also noted that this kind of jump can easily be found in long-term eddies, but we do not check it systematically (our time are spent to the paper itself).

Comment L343 "A similarity vector": is it obvious if the number of missing eddies is reduced due to the fact that it is a vector or your approach of using the overlapping of territories?

**Reply**: Yes, when we use overlapping of territories, the missing eddies is reduced.

Comment L345 "as scale": "a scalar"? Btw, oftentimes, people previously have used not only distance but in addition a similarity parameter as well, but a scalar defined based on eddy properties such as amplitude/size/vorticity etc.

**Reply**: suggestion followed.

Comment L364 "that cyclonic": "a pair of cyclonic"?

**Reply**: suggestion followed.

Comment L365 "for cyclones": "for atmospheric cyclones"?

**Reply**: suggestion followed.

Comment L369: I would be interested in the number of merging/splitting events per lifetime, could you mention it? And/or if you showed Fig 9 in terms of eddy tracks passing a grid box per year (instead of eddy extrema), one could easily compare Fig 12 to Fig 9 to roughly estimate such a number.

**Reply**: We add Figure 13 to show this. We have redraw the figure.

Comment L388: I suggest to highlight the regions you mention in the Figure, with a box or so.

**Reply**: We add a blue box for NEC and black box for eddy desert.

Comment L407 "false": I don't like "false" as it is not obvious what extrema are true and what are false. The original data may have errors and hence false extrema, the smoothed data likewise.

**Reply**: we modify it to "additional"

Comment L411 "This is one of the reasons why we need look-ahead": "The ambiguity of the eddy detection procedure and the potential errors in the input data strongly suggest the application of a look-ahead approach. The resulting eddy tracks are largely insensitive for instance to a filtering method applied to the input SLA data." Or so.

**Reply**: We modified the text as suggestion.

Comment L413: in this section, it would be nice to quantify a bit more the uncertainty due to variations of N and variations of rc. For N you have partly done it (L424 where you provide the L418 "The numbers of eddies seldom change": "The number of eddies does not change substantially". Can you quantify this a bit more? E.g. provide the L421 "The numbers of merging and splitting events seem to converge for rc >0.5 as N increases.": "The sensitivity of the number of merging and splitting events seem to converge to 0 for rc >0.5". Can you briefly comment on why this is the case?

**Reply**: If rc<0.5, there might be two eddies which are taken as successors of an given eddy at

seem time. So the tracks might be randomly jump from one eddy to another. The results are not believable. We add this comment.

Comment L425 "missing eddies, which may also reduce both total eddy numbers and dynamic events.": I would have expected the opposite (unless you consider only eddies with a certain minimum lifespan, e.g. 30 days. I guess this is the case here?)

**Reply**: Yes, it depends on which one would be counted as eddy. For eddies with lifetime>30 days, the missing eddies may reduce the total numbers. But if one consider all eddies (even lifetime>one day), the missing eddies increase the total numbers.

Comment L427 "0.6-0.7": why is this optimal?

**Reply**: In our programs, it is reasonable. In one hand, we first require that rc>0.5. On the other hand, we know there is area error in calculation (~10%) since only eddy grids are taken into consider. This is also the reason why we need rc<0.9 or even smaller. So the optimal value should be within 0.5-0.9, and ~0.7 is just in this middle.

Comment L445 "Thus, it is difficult to directly compare the influences of eddy territory using different tracking algorithms": I don't see how this follows from what you said in the previous sentences. Also, you originally point out that you need specifically identified eddies which allow you to find the merging and splitting events. Which is why you use the watershed algorithm. Hence, overall I don't really see the point of this paragraph.

**Reply**: The point is that it is difficult to directly compare the influences of eddy territory using different tracking algorithms. We rewrite this paragraph and move it behind.

Comment L447: The paragraph is difficult to follow except for the last two sentences. I like the sensitivity test with respect to the minimum amplitude, only an issue is that you don't strictly test the sensitivity of the eddy boundary as the modification of the amplitude threshold also changes the number of identified eddies (as you state yourself). The whole point of section 5.3 appears to be that the sensitivity of the eddy boundary is not straightforward to determine but is estimated to be small.

**Reply**: Yes, the main point is that the sensitivity of the eddy boundary is not straightforward to determine but is estimated to be small for present eddy identification.

Comment L457: does it make sense at all to test the tracking algorithms with the input from other detection algorithms which do not include what you refer to with "segmentation"?

**Reply**: We hope the newly additional paragraph in section 5.4 answers the question.

Comment L465 "A better way to obtain these parameters might be to use a nonlinear fitting of the flow field": unclear, please rephrase. The point of the whole paragraph is unclear to me.

**Reply**: Since they may not be sufficiently accurate for some applications. Besides, some physical quantity (circulation, angular momentum, energy) are required to be accurately calculated in the investigation of eddy dynamics process. A better way to obtain these characteristics might be to use a nonlinear fitting of the flow field with appropriate models other than simply estimated from identification.

Comment L472 "computation": is this referring to the computation time of the tracking procedure only or including the identification? This is great. Could you repeat the detail here on the model domain and other constraints (eddy size, amplitude, N etc) that the reader can get an impression what it would mean for his/her own application?

**Reply**: In a personal computer with CPU of i7-6700k and 4.00 GHz, program

Mei.f90 uses 15 minutes to identify *vortices* (snapshot of eddy), program Vortex.f90 uses about 20 minutes to establish similarity, and program Eddy.f90 uses about 10 minutes to track eddies in the North Pacific Ocean (NPO).

Comment L475 ff: I would stress this, this is the really exiting part for me as an oceanographer.

Reply: (i) the strong eddy interaction which leads to genesis and termination of eddies (ii) the weak eddy interaction which associates with merging/splitting events (iii) the weak eddy interaction which modulates the eddy track and motion.

*Figures:*

*General:*

-If you show snapshots of eddies, e.g. Fig 11 or 13, I suggest to roughly maintain an aspect ratio which doesn't distort the eddies that much (the extreme case is Fig 13!).

-please describe everything you show in your Figures, e.g. "white dots mark eddy centers" etc.; I mention some of the missing references below.

-the resolution could be increased in some of the figure and the labels increased (e.g. longitudes/latitudes are sometimes difficult to read).

Fig. 1: -mention that the background field shows SLA, and white dots mark eddy centers.

**Reply**: we add the notation.

Fig. 2: -say what the letters h, A, t etc in the figure refer to.

**Reply**: The h represents background SLA value, A represents amplitude of eddy, and t represents the map at different time.

Fig. 3: -cryptic caption, please expand (explain letters, what quantity do you illustrate in panel a etc).

**Reply**: Watershed as the natural division of eddies C1 and C2. from top view, where contours represent SLA.  (b) The particles P1 and P2 on the watershed flow downward to the eddy centres C1 and C2 from cross-section view.

Fig. 5: -missing colorbar, say what A1 etc refer to.

**Reply**: Redraw the figures, and add the reference.

Fig. 6a: I suggest to shade only the overlap/intersection of eddy territories.

Fig. 6b:

-change "their are four types" to "there are four similarity types".

-in the text it says you used rc=2/3; you may want to illustrate this threshold here.

-it would help if you noted the "types of similarity" you write about in the text in the Figure in plain English, i.e. "unrelated" etc. You could also insert E2 to E4 in the respective types/quadrants to continue the illustration of 6a.

**Reply**: We redrew the figure, and modified the text.

Fig. 7:

-why are the eddies labelled now Ec1 etc instead of E1 etc? If possible, make it consistent.

-cryptic caption, please expand.

**Reply**: We expand them. (a) Three typical cases of successors (T1, T2 and T3) from one day (day 0) to another (day 1). (b) The eddy at day 0 may have different successors corresponding to different numbers of "look-ahead" days, e.g., Ed1 at day 0 may have a T3 eddy on day 2, and have two T2 eddies on day 3.

Fig. 8b:

-"of the eddies" or of an example eddy?

**Reply**: modified

Fig. 9:

-are all extrema counted here or only the ones belonging to eddies which existed at least 30 days?

-is the scale logarithmic or just nonlinear?

-mention the red box.

-instead of number of eddy extrema you could show number of eddy tracks per year (i.e. count an eddy which passes through only once – or have you done this? This is unclear) to make the numbers interpretable together with Fig 12.

-9c: I am a bit surprised that the eddy numbers are distributed that evenly.

**Reply**: Yes, they are 30 days at least. And the scale is logarithmic. We add the notation of red boxes. At least the example in Fig.11 implies that this region is anticyclonic eddies dominated.

Fig. 12:

-adjust colorbar so that the figure does not appear only in blueish colors.

-I cannot see the spatial pattern you describe in the text very clearly.

-12d: you could show the ratio of merging and splitting to highlight the difference (as you have done in Fig 9 for cyclones/anticyclones).

**Reply**: We redraw the figures.

Fig. 13:

-note in the caption what are the box, letters, white dots.

**Reply**: We add them.

Fig. 14a: I like figure 14 a lot, it is great that you tested the sensitivity of the results to the parameters.

-why do you have the spike at rc=0.6?

**Reply**: It is a typo in data, the new figure does not have this spike.

---

## Author Response (AR2)

(Abstract)

Line 14. "rates" -> "values"?

**Reply**: We modify to ratios of overlap area between two eddies to the area of each eddy.

Line 15. "temporarily" (spelling).

**Reply**: suggestion followed.

Lines 19-20. ".. the number N of look-ahead time steps, and .."

**Reply**: suggestion followed.

Line 20. I Googled "computational complexity". I think the number of operations (which is what you count) is one aspect of computational complexity but might be better described as number of operations.

**Reply**: Yes, computational complexity here refers to time complexity, i.e., the number of operations. We now used *time complexity* hereafter. The computational time of the algorithm is detailed in section 5.4.

(1 Introduction)

Lines 55, 426, 430 (and maybe other places) "noise" not "noises".

**Reply**: thank you.

Line 60. Something missing at ".. is [Chelton .."

**Reply**: we modified to '.. it is a solution to the "missing eddy'' problem [Chelton ...'

Line 63. Omit "following"

**Reply**: suggestion followed.

(2.1 Input data)

Line 118. "detection, .."

**Reply**: thank you.

Line 120. Omit "to"

**Reply**: suggestion followed.

Line 127. "territories" -> "areas". [You said you would use "areas". "territory" suggests land, not sea. Especially, if ".. and boundaries" then "areas" is the word. "areas and boundaries" might be "territories" on land; "domains" is the mathematical word.

**Reply**: suggestion followed.

(3.1 Overview of GEM)

Line 205. ".. In addition to eddy area and boundary, .. input, the critical values of area ratio rc and N. See section 5.2 ..." (duplication of "input").

**Reply**: suggestion followed.

(3.2 Map link)

Line 210. Maybe "territory" -> "domain"

**Reply**: suggestion followed.

Lines 217, 218, 219, 226. Maybe likewise "territories" -> "domains". For all these "area" is also possible but less precise.

**Reply**: suggestion followed.

Line 221. "B2 only had partial similarity". B1 is discussed but not B2. Also not obvious in figure.

**Reply**: We had used 19970328 and 19970329 in the first time (and the description text was also for such dates), but incorrectly used 19970301 and 19970302 in the last time. Now we redraw the figures with 19970328 and 19970329.

Lines 237-238. I think you need to say here, how the values of r1 and r2 decide whether it is T1 or T2. See lines 262-263 for how r2 determines T1; this should be at line 238.

**Reply**: suggestion followed.

Line 274. "(section 3.4)"

**Reply**: thank you.

Line 275. ". . similar, and 2) the earliest day. Rule 1 . ." (i.e. omit "first" twice, it is confusing).

**Reply**: suggestion followed.

Lines 276, 282. "E1" -> "Ed1".

**Reply**: suggestion followed.

Line 283. ". . as the following Ed2, Ed3 respectively." ?

**Reply**: suggestion followed.

(3.3 Track tree)

Lines 304, 305. Again you should say how the values of r1 and r2 decide whether it is "first type" or "second type".

**Reply**: criteria have been added.

(4.1 Eddy tracks)

Lines 356-357. "Eddy C1 . . trajectory is the longest". This is a good reason for choosing it as the only example; it has the best chance of showing any problem! You should say this and also that you did actually look at many others.

**Reply**: Yes! We pointed out that in the above paragraph.

Line 371. "it" – not clear what this refers to.

**Reply**: Sorry for that. We modified "it" to "the application of similarity vector".

(5.2 Impact of variations of parameters)

Lines 447-448. Maybe better "To discuss the impact of the GEM critical value rc and look-ahead time N, we carry out . . 2012. The number of eddies"

**Reply**: suggestion followed.

Line 474. "O-W" needs defining here (bring definition from lines 498-9).

**Reply**: suggestion followed.

Line 474 "sensitive". What is sensitive to what? I think you perhaps mean that "suitable rc depends on whether it is O-W or SLA identification".

**Reply**: Yes. GEM based on Okubo–Weiss (O–W) parameter identification O-W based identification is much sensitive to the critical value rc.

Line 481. "to though". This does not make sense and I cannot say what you do mean!

**Reply**: Sorry for that. It has been modified

Lines 496-497. Better "comparison of eddies identified by different methods, since the eddy detection algorithms differ a lot from each other." ?

**Reply**: suggestion followed.

Figure 4. This is OK on my screen but had dark infill when I printed it; please check.

**Reply**: We redraw the figure with light colors.

Figure 6. There is no illustration of T1.

**Reply**: Yes, there is no T1 in the figure, since there is at most one eddy that can be marked as a T1 (merging) or T3 (living) eddy on the following day (section 3.2.2). We add this notation to the label of figure.

Line 692. ". . (b) There are four similarity types . ."

**Reply**: suggestion followed.

[revised manuscript text omitted]

---

## Author Response (AR3)

(Abstract)

Line 11. "The Genealogical . . (GEM) presented here is an efficient . ."

**Reply**: thank you.

Lines 16-17. ". . Second, for tracking when an eddy splits, GEM uses both "parent" (the original eddy) and "child" (eddy split from parent) and the dynamic processes . ."

**Reply**: thank you.

Line 18. "Additionally, a new look-ahead approach . ."

**Reply**: thank you.

Line 21. I do not like "time complexity" either! I think you just mean "number of computing operations" which can be more simply stated as "computer time" or "runtime" as on line 103 [the unit of time would be the time for one computer operation]. See also lines 68, 96(twice), 103, 273, 320, 322, 324, 325, 327, 329, 549 (and perhaps other places).

**Reply**: we modified to "computer time".

(Introduction)

Line 70. ". . motion of an eddy. . ."

**Reply**: thank you.

Line 82. ". . risk of missing an eddy . ."

**Reply**: thank you.

Line 84. ". . may arise from two or more eddies (at time step k-1)"

**Reply**: thank you.

Line 97. ". . where L denotes the number of pixels . ."

**Reply**: thank you.

Line 103. "runtime. Results . ."

**Reply**: thank you.

(2.2 Eddy identification)

Line 123. ". . identified as multinuclear . ."

**Reply**: thank you.

Line 150. I think cyclonic eddies have lowered sea-surface, i.e. SLA < 0, etc. So ". . SLA < -3 cm for cyclonic eddies and SLA > 3 cm for anticyclonic . ."

**Reply**:suggestion followed.

Line 153. Omit "those"

**Reply**: thank you.

(2.3 Eddy segmentation . .)

Line 156. ". . eddies. Different"

**Reply**:thank you.

(3.2.1 Eddy similarity)

Lines 233-234. ". . When we apply GEM . . Pacific Ocean (section 4.1), we choose . ."

**Reply**: thank you.

Lines 235-242. In figure 6(a) the subscripts (1,2,3,4) in E1, . . E4 are very small. I suggest you use E1, E2, E3, E4 (no subscripts) in the figure and in the text here.

**Reply**: suggestion followed.

Lines 245-249 and Figure 6a. The description seems to correspond to rc about 0.4 or less. E.g. S12/S1 <½ and with rc = 2/3 this would be case T2 not T3. Also S13 ~ ½ and with rc = 2/3 this would be case T0 not T2. You might find it easier to say rc = 0.3 than to re-draw the figure but re-drawing for rc = 2/3 would be less confusing for the reader.

**Reply**: figure 6b is redrawn now.

Lines 246-247. "..different relationship ("splitting", marked as T2 in Fig. 6b)..."

**Reply**:thank you.

(3.2.2 Eddy Look-ahead)

Lines 259-260. "..procedure. Examples for a given eddy.."

**Reply**: thank you.

Line 263. "..In the middle row, eddy Ec1 has two"

**Reply**: thank you.

Line 265. "In the lower row, eddy Ec1 has T2..and T3..(respectively Ec2, Ec3)". [E1 -> Ec1] But in the figure Ec2 and Ec3 are the other way round! Better to change the figure so Ec2 <-> T2 and Ec3 <-> T3.

**Reply**: suggestion followed, figure 7a is redrawn.

(3.3.1 Eddy branch)

Lines 299-300. "..high for the record of eddy C to be appended to either parent. There might be.."

**Reply**:thank you.

(3.4 Runtime (?!))

**Reply**: we modified to "computer time".

Line 329. "..much less than.."

**Reply**:thank you.

(4.1 Eddy tracks)

Line 362. "trajectory from.."

**Reply**: thank you.

Line 371. "..application of our similarity vector.."

**Reply**: thank you.

Line 373. "..application of the similarity vector.."

**Reply**: thank you.

Line 374. "..as a scalar.."

**Reply**: thank you.

(4.2 Eddy merging and splitting)

Line 386. "..(Fig. 5).."

**Reply**: thank you.

Line 387. "..location south of B1 and B2.."

**Reply**: thank you.

Lines 388-389. "..After that, eddy B2 merged into eddy B3..eddies B1 and B3 eventually merged.."

**Reply**: thank you.

Line 395. Omit "which is". [A general point; "which" must follow immediately after the item it refers to. So here the meaning is clearer without "which".]

**Reply**: thank you.

Line 397. ". . eddies (A1, A2; B1, B2, B3) are anticyclonic, they have clockwise rotation and orbits, like point vortices"

**Reply**: thank you.

(4.3 Census)

Line 403. ". . Figure 12 and are similar . ."

**Reply**: thank you.

Lines 417-418. I do not see "high frequency of merging and splitting" in the "eddy desert" in Figure 12. It might just be "high frequency" relative to the number of eddies but that is not enough to explain the lack of eddies.

**Reply**: The present figures lost such information as we draw absolute frequency of merging/splitting events by following reviewers suggestion. However, the previous figures (removed by review comment) have such information as we drew relative frequency of merging/splitting events. We append this figure as supplement.

[Figure]

The frequency of dynamic al processes, normalized by the number of eddy. (a) The merging frequency for cyclonic eddies. (b) The splitting frequency for cyclonic eddies.

Lines 419-420. ". . may be due to eddies being too small to be detected or to eddy lifetimes being too short . ." [You cannot have a "fact" about eddies that you do not know about.]

**Reply**: suggestion followed. It has been modified.

Lines 423-425. ". . regardless of eddy polarizations . . splitting events increase approximately linearly with eddy lifetime. However, merging and splitting events are more frequent for anticyclonic eddies than for cyclonic eddies." Please note that there is duplication with the figure caption lines 744-745. I think this is better here because it is interpretation not explanation of the figure.

**Reply**: suggestion followed.

(5.1 Data noise)

Line 435. "As a sensitivity test . ."

**Reply**: thank you.

Line 441. ". . even when it is small. It might . ."

**Reply**: thank you.

(5.2 Impact of variations . .)

Lines 456-457. These numbers of splitting and merging events do not agree with figure 15b. Perhaps 121220 and 120612 respectively?

**Reply**: Sorry for that. It is a typo in data. We modified it.

Lines 465-466. ". . reasonable. On the one hand, . . .we know there is error (~10%) in calculating eddy area since . .consideration. . ."

**Reply**: suggestion followed.

Lines 469-470. ". . N=1 and N=2 have respectively 95.5% and 98% . .for N=4. To reduce . ."

**Reply**: suggestion followed.

Line 473. Omit "of"

**Reply**: thank you.

Line 474. ". . independent of MEI . ."

**Reply**: thank you.

Line 475. ". . in eddy identification. . ."

**Reply**: thank you.

Lines 475-476. ". . is more sensitive . . than is SLA-based GEM, since . ."

**Reply**: thank you.

Line 477. ". . independent of N, . ."

**Reply**: thank you.

Line 478. ". . too large, eddies may move too far so that eddy"

**Reply**: thank you.

Line 479. "constraint"

**Reply**: thank you.

(5.3 Impact of eddy boundary)

Lines 488-491. "Different identification methods may give different eddy boundaries, although the eddy centre is relatively robust. Eddy area S is sensitive to eddy boundary but it is difficult to compare directly the influence of eddy boundary differences that result from identification method choice. However, the area ratio . ." [I have rearranged these lines. I have also omitted the sentence about "indirect way"; it does not seem to help.]

**Reply**: thank you.

Line 497. "However, such a sensitivity test . . values used in the same"

**Reply**: thank you.

Line 505. ". . eddy boundary differences resulting from using . ."

**Reply**:thank you.

(5.4 Future research)

Line 513. "Eddies identified by using algorithms without watershed segmentation can also be tracked .."

**Reply**: thank you.

Lines 516-517. ". . because most merging/splitting occurred between eddies more than a certain distance apart. This weak interaction . ." [I am not sure about "more" but assume this from "weak" interaction. If the eddies need to be close for merging/splitting then "less" not "more"].

**Reply**: we didn't use "more than.."

516   some far distance could still be recorded, because most of merging/splitting records occurred at the interaction of
517   two eddies with a certain distance. This weak interaction still can't be recorded by previously interaction-free

Line 524. "..physical quantities (.."
**Reply**: thank you.
Line 527. "..other than simple estimation from identification."
**Reply**: thank you.
Line 539. Omit "on those problems"
**Reply**: thank you.
Lines 551-552. "..and in "eddy deserts"..." This was not obvious to me (seem comment on lines 417-418).
**Reply**: thank you.

(Figures and captions)
Line 664. July 26 to August 3.
**Reply**: thank you.
Figure 4. In figure, "territory" -> "domain"
**Reply**: figure redrawn
Figure 5b. Add "B2" eddy label.
**Reply**: B2 is added
Figure 6a. See comment on lines 235-242. The subscripts (1,2,3,4) in E1, ..E4 are very small. I suggest you use E1, E2, E3, E4 (no subscripts) in the figure here and in the text.
**Reply**: figure is added
Lines 744-745. See comment on lines 423-425. Repetition! I think these two sentences are interpretation, should be in the text and are not needed here.
**Reply**: removed.